# Test-time Contrastive Concepts for Open-world Semantic Segmentation

## Abstract

Recent CLIP-like Vision-Language Models (VLMs), pre-trained on large amounts of image-text pairs to align both modalities with a simple contrastive objective, have paved the way to open-vocabulary semantic segmentation. Given an arbitrary set of textual queries, image pixels are assigned the closest query in feature space. However, this works well when a user exhaustively lists all possible visual concepts in an image, which contrast against each other for the assignment. This corresponds to the current evaluation setup in the literature which relies on having access to a list of in-domain relevant concepts, typically classes of a benchmark dataset. Here, we consider the more challenging (and realistic) scenario of segmenting a single concept, given a textual prompt and nothing else. To achieve good results, besides contrasting with the generic "background" text, we propose two different approaches to automatically generate, at test time, textual contrastive concepts that are query-specific. We do so by leveraging the distribution of text in the VLM's training set or crafted LLM prompts. We also propose a metric designed to evaluate this scenario and show the relevance of our approach on commonly used datasets.

## 1 Introduction

Vision-language models (VLMs) such as CLIP (Radford et al., 2021) are trained to align text and global image representations. Recently, VLMs have been proposed for denser tasks (Zhou et al., 2022; Ghiasi et al., 2022; Li et al., 2022). This includes the challenging pixel-level task of open-vocabulary semantic segmentation (OVSS), which consists of segmenting arbitrary *visual concepts* in images, i.e., visual entities such as objects, stuff (e.g., grass), or visual phenomena (e.g., sky). To that end, several methods exploit a frozen CLIP model with additional operations (Zhou et al., 2022; Bousselham et al., 2024; Wysoczańska et al., 2024b;a), or fine-tune the model with specific losses (Xu et al., 2022; Ranasinghe et al., 2023; Cha et al., 2023; Luo et al., 2023; Mukhoti et al., 2023).

Most OVSS methods label each pixel with the most probable prompt (or query) among a finite set of prompts provided as input, contrasting concepts with each other. This works well for benchmarks that provide a large and nearly exhaustive list of things that can be found in the dataset images, such as ADE20K (Zhou et al., 2019) or COCO-Stuff (Caesar et al., 2018). However, when given a limited list of queries, these methods are bound to occasionally suffer from hallucinations (Wysoczańska et al., 2024b; Miller et al., 2024). In particular, common setups do not handle the case where only a single concept is queried (Cha et al., 2023; Xu et al., 2022), which results in classifying all pixels using the same concept.

To catch such hallucinations, a common strategy consists in using an extra class labeled 'background', intended to capture pixels that do not correspond to any visual concept being queried. This extra class is already present in object-centric datasets, such as Pascal VOC (Everingham et al., 2012). It provides an easy, generic concept to be used as a negative query, i.e., to be used to contrast with actual (positive) queries, but to be discarded from the final segmentation. However, the notion of background is not well defined as it is context-dependent, therefore providing suboptimal contrasts. This strategy also fails when a queried concept (e.g., "tree") falls in the learned background (which commonly encompasses trees).

In this work, we consider the practical and realistic OVSS task in which only one or a few arbitrary concepts are to be segmented, leaving out the remaining pixels without any prior knowledge of other concepts that may occur in an image. We name this setup *open-world* Given a query, instead of

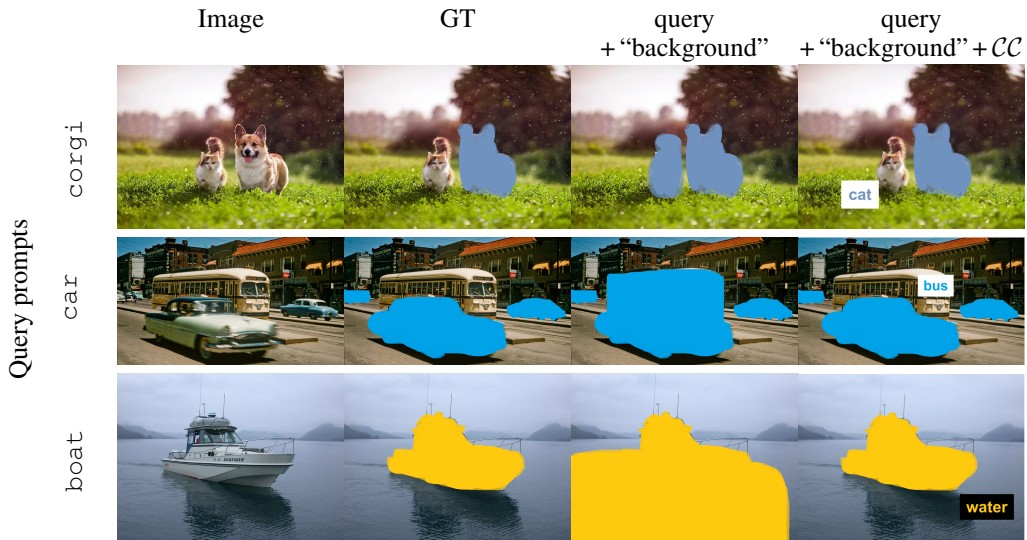

Figure 1: **Illustration of our proposed open-world scenario and benefits of contrastive concepts** ($\mathcal{CC}$). We investigate open-world segmentation, where only one (or a few) visual concepts are to be segmented (2nd column), while all concepts that can occur in an image are unknown. Contrasting the query with "background" allows us to obtain a coarse segmentation (Ranasinghe et al., 2023; Wysoczańska et al., 2024b) (3rd column), but is not enough to catch all pixels *not* corresponding to the query when they are related or co-occur frequently in the VLM training set. Our automatically-generated *contrastive concepts* ($\mathcal{CC}$) (4th column) help to separate and disentangle pixels of the query (right column, generated $\mathcal{CC}$ in text boxes), therefore achieving better segmentation.

assuming access to a dataset-specific set of classes (a closed-world setup), we propose to automatically suggest contrastive concepts that are useful to better localize the queried concept, although they can later be ignored. In particular, we focus on predicting concepts likely to co-occur with the queried concept, e.g., "water" for the query "boat" (as visible in Fig. 1), thus leading to better segment boundaries when prompted together.

Moreover, we argue that this scenario needs to be evaluated to better understand the limitations of open-vocabulary segmentation methods. We therefore propose a new metric to measure such an ability, namely IoU-single, which considers one query prompt at a time and thus does not rely on the knowledge of potential domain classes.

To summarize, our contributions are as follows:

- We introduce the notion of test-time contrastive concepts and discuss the importance of contrastive concepts in open-vocabulary semantic segmentation.
- We analyze the usage of "background" as a test-time contrastive concept, which has been so far accepted but not discussed.
- We propose a new single-query evaluation setup for open-world semantic segmentation that does not rely on any domain knowledge. We also propose a new metric to evaluate the grounding of visual concepts.
- We propose two different methods to generate test-time contrastive concepts automatically and show that our approaches consistently improve the results of 7 different popular OVSS methods or backbones.

## 2 RELATED WORK

**Open-vocabulary semantic segmentation.** VLMs trained on web-collected data to produce aligned image-text representations (Radford et al., 2021; Jia et al., 2021; Zhai et al., 2023) had a major impact on open-vocabulary perception tasks and opened up new avenues for research and practical applications. While CLIP can be used *off-the-shelf* for image classification in different settings, it does not produce dense pixel-level features and predictions, due to its final global attentive-pooling

(Zhou et al., 2022; Jatavallabhula et al., 2023). To mitigate this and produce dense image-text features, several methods finetune CLIP with dense supervision. Other approaches devise new CLIP-like models trained from scratch using a pooling compatible with segmentation. Their supervision comes from large datasets annotated with coarse captions (Ghiasi et al., 2022; Ranasinghe et al., 2023; Liang et al., 2023; Xu et al., 2022; Liu et al., 2022; Xu et al., 2023; Mukhoti et al., 2023; Cha et al., 2023), object masks (Rao et al., 2022; Ghiasi et al., 2022; Ding et al., 2022) or pixel labels (Li et al., 2022; Liang et al., 2023). However, when models are finetuned, they face feature degradation (Jatavallabhula et al., 2023), or require long training cycles on huge amounts of images when trained from scratch.

CLIP densification methods have emerged as a low-cost alternative to produce pixel-level image-text features while keeping CLIP frozen (Zhou et al., 2022; Wysoczańska et al., 2024a; Jatavallabhula et al., 2023; Abdelreheem et al., 2023; Wysoczańska et al., 2024b; Bousselham et al., 2024). The seminal MaskCLIP (Zhou et al., 2022) mimics the global pooling layer of CLIP with a $1 \times 1$ conv layer. The aggregation of features from multiple views and crops (Abdelreheem et al., 2023; Kerr et al., 2023; Wysoczańska et al., 2024a; Jatavallabhula et al., 2023) also leads to dense features, yet with the additional cost of multiple forward passes. Some methods (Shin et al., 2022; 2023; Karazija et al., 2023) rely on codebooks of visual prototypes per concept, including per-dataset negative prototypes (Karazija et al., 2023), or leverage self-self attention to create groups of similar tokens (Bousselham et al., 2024). The recent CLIP-DINOiser (Wysoczańska et al., 2024b) improves MaskCLIP features with limited computational overhead thanks to a guided pooling strategy that leverages the correlation information from DINO features (Caron et al., 2021).

**Prompt augmentation.** Prompt engineering is a common practice for adapting Large Language Models (LLMs) to different language tasks (Kojima et al., 2022) without updating parameters. This strategy of carefully selecting task-specific prompts also improves the performance of VLMs. For instance, in the original CLIP work (Radford et al., 2021), dataset-specific prompt templates, e.g., "a photo of the nice $\{\cdots\}$" were devised towards improving zero-shot prediction performance. Although effective, manual prompting can be a laborious task, as templates must be adapted per dataset and sufficiently general to apply to all classes. Afterwards, different automated strategies were subsequently explored, e.g., scoring and ensembling predictions from multiple prompts (Allingham et al., 2023). Prompts can also be augmented by exploiting semantic relations between concepts defined in WordNet (Fellbaum, 1998) to generate new coarse/fine-grained (Ge et al., 2023) or synonym (Lin et al., 2023) prompts. LLMs can be used as a knowledge base to produce rich visual descriptions adapted for each class starting from simple class names (Pratt et al., 2023; Menon & Vondrick, 2023). Prompt features can be learned by considering visual co-occurrences (Gupta et al., 2019), a connection between training and test distributions (Xiao et al., 2024), mining important features for the VLM (Esfandiarpoor et al., 2024) or by test-time tuning on a sample (Shu et al., 2022). Most of these strategies have been designed and evaluated for the image classification task, and their generalization and scalability for semantic segmentation are not always trivial. Here, we aim to obtain better prompts for semantic segmentation to separate queried object pixels from their background. We do this automatically without supervision and without changing the parameters of either the text encoder or the image encoder, leveraging statistics from VLM training data or LLM-based knowledge.

**Dealing with contrastive concepts in OVSS.** Our contrastive concept discovery is tightly related to *background handling* in the context of open-vocabulary semantic segmentation, since the standard benchmark datasets for this task, originally designed for supervised learning, use *background* to describe unlabeled pixels, for example, to cover concepts outside of the dataset vocabulary. There are three main types of approaches to address this problem. The first one is to threshold uncertain predictions (Cha et al., 2023; Bousselham et al., 2024; Xu et al., 2022) with a given probability value (Xu et al., 2022; Bousselham et al., 2024) or clip similarities (Cha et al., 2023). The second group of methods leverages the object-centric nature of certain datasets by defining background through visual saliency (Wysoczańska et al., 2024a;b). Finally, a significant body of work addresses the same issue by defining dataset-level concepts either by adding handcrafted names of concepts to the background definition (Lin et al., 2024; Yu et al., 2023; Ranasinghe et al., 2023; Cho et al., 2024) or by extracting visual *negative prototypes* with a large diffusion model (Karazija et al., 2023). In contrast, in this work, we aim for automatic discovery of contrastive concepts without any prior access to the vocabulary used for the annotation of the dataset.

**Visual grounding** is the task of localizing within images specific objects from text descriptions. The major instances of visual grounding tasks are *referring segmentation* that produce pixel-level predictions for one (Hu et al., 2016; Ding et al., 2023; Wang et al., 2022) or multiple target objects (Liu et al., 2023) given a text description, and *referring expression comprehension* (Chen et al., 2018; Deng et al., 2021; Liao et al., 2020; Liu et al., 2024) that detects objects. Similarly to referring segmentation, we aim to segment specific user-defined objects. In contrast, we do not use supervision to align textual descriptions with object masks and do not focus on text-described relations between objects and mine contrastive concepts to disentangle target objects from the background.

## 3 OPEN-WORLD SEGMENTATION WITH TEST-TIME CONTRASTIVE CONCEPTS

We consider the following segmentation task: given an image and a set of textual queries characterizing different visual concepts, the goal is to label all pixels in the image corresponding to each concept, leaving out unrelated pixels, if any. Moreover, we want to do so without any prior knowledge of what concepts could be prompted at the test time. That is, not only do we want to be *open-vocabulary* in terms of the choice of words for querying, but we also want to be *open-world*, that is, not specialized in a given domain or set of categories. For evaluation purposes, segmenting a specific dataset thus shall not assume anything about the dataset, such as knowledge of represented classes.

### 3.1 INTRODUCING THE USE OF TEST-TIME CONTRASTIVE CONCEPTS

**Closed-world vs open-world open-vocabulary semantic segmentation.** Even when it is open-vocabulary, traditional semantic segmentation is *closed-world* in the following sense. Given an RGB image $\mathbf{I} \in \mathbb{R}^{H \times W \times 3}$ and a set of textual queries $q \in Q$, semantic segmentation yields a map $\mathbf{S}_{\text{closew}} : \{1...H\} \times \{1...W\} \mapsto Q$, where each image pixel has to be assigned one of the queries as a label. In contrast, *open-world* segmentation considers an extra dummy label '$\perp$' to represent any visual concept that is different from the queries. The segmentation map, in this case, is then $\mathbf{S}_{\text{openw}} : \{1...H\} \times \{1...W\} \mapsto Q \cup \{\perp\}$. For instance, to label a boat, it is enough to ask for the "boat" segment; other pixels (sky, sea, sand, rocks, trees, swimmers, etc.) are expected to be labeled $\perp$ and thus ignored.

Following, we show how to use any open-vocabulary segmenter in an open-world fashion. We only assume that the segmenter uses a CLIP-like architecture with a text encoder, noted $\phi_{\text{T}}(\cdot)$, used to extract textual features $\phi_{\text{T}}(q) \in \mathbb{R}^d$ for any query $q$, where $d$ is the feature dimension. Patch-level features $\phi_{\text{V}}(\mathbf{I}) \in \mathbb{R}^{h \times w \times d}$ are generated using the visual encoder, noted $\phi_{\text{V}}(\cdot)$, where $h = H/P$, $w = W/P$, and $P$ is the patch size. The cosine similarities between each query feature and a patch feature are then used as logits when upsampling to obtain pixel-level predictions. It yields a closed-world segmentation, given our definition above.

From such segmentation, open-world segmentation could be derived by assigning a pixel (or patch) to a query if the cosine similarity between the visual embedding and the query embedding is above a given threshold. However, in practice, it has been commonly observed that the CLIP space is not easily separable (Miller et al., 2024), thus making the definition of such a threshold difficult without overfitting the query or datasets (Bousselham et al., 2024; Cha et al., 2023).

**Train-time contrastive concepts.** Cues to separate visual concepts without supervision primarily come from data where these concepts occur separately and are described in their captions. If some concepts always co-occur, they are harder to be told apart. This applies in particular to OVSS models that are trained only from captioned images, rather than from dense information. Sharing a caption pushes their embedding to align on a common textual feature, which in turn tends to bring the visual embeddings closer together. Still, such frequently co-occurring visual concepts can often be separated in a closed-world setting: pixels (or patches) are then just mapped to the query with which they align the most. However, a problem arises if a visual concept of a query $q$ can be mistaken for another visual concept present in the image but not queried (e.g., querying "boat" but not "water" as in Fig. 1).

**Test-time contrastive concepts.** To address this problem, we propose to use one or more additional textual queries of visual concepts that are likely to contrast well with $q$. For instance, when querying "boat", we want to add the query "water". We name such queries *test-time contrastive concepts* and note them $\mathcal{CC}_q$. We further propose different solutions to automatically generate $\mathcal{CC}_q$, and such

without assuming prior access to the image domain. Given prompt queries $\{q\} \cup CC_q$, we perform closed-world segmentation and assign to the dummy label $\perp$ any patches which are labeled as $CC_q$.

**Multi-query segmentation.** This principle can be generalized to several simultaneous queries $Q$, with $|Q| > 1$, considering the union of their contrastive concepts $CC_Q = \bigcup_{q \in Q} CC_q$. Open-world multi-query segmentation consists in segmenting $Q \cup CC_Q$, and ignoring pixels not assigned to the queries in $Q$, as in the single-query case. However, some queries in $Q$ may already contrast each other, which puts them in competition with the set of contrastive concepts $CC_Q$ and could lead to their elimination when pixels labeled in $CC_Q$ are discarded. To prevent it, we propose to exclude contrastive concepts $CC_Q$ that are too similar to queries $Q$, e.g., with a cosine similarity of text features above some threshold $\beta$: $CC_Q = \bigcup_{q \in Q} \{q' \in CC_q \mid \phi_T(q') \cdot \phi_T(q) \leq \beta\}$. In the following, for simplicity, we only consider the single-query scenario, where $|Q| = 1$.

Moreover, to the best of our knowledge, none of the currently used evaluation benchmarks for OVSS allows us to measure the effectiveness of such $CC$s. We, therefore, propose a variant of the traditional evaluation metric for semantic segmentation and discuss it in detail in Sec. 4.1.

## 3.2 CONTRASTING WITH "BACKGROUND" ($CC^{BG}$)

In recent work (Ranasinghe et al., 2023; Wysoczańska et al., 2024a;b), the word "background" has been used to try to capture a generic visual concept to help segment foreground objects, separating them from their background. In our framework, it amounts to defining "background" as a test-time contrastive concept to any query $q$. In other words, it defines $CC_q^{BG} = \{\text{"background"}\}$.

However, if the word "background" feels natural to us, it is not obvious why it should also make sense in the CLIP space. In fact, this formulation is not contextual, meaning that the contrastive concept is not specific to the query, which might be suboptimal. Worse, the "background" samples that CLIP learned from could accidentally include the visual concept of the query, which could make the query representation close to the background representation and defeat the contrast mechanism.

To sort it out, we investigate the occurrence of "background" in VLM training data. First, we use the metadata provided by Udandarao et al. (2024), which describes the representation of four thousand common concepts in LAION-400M (Schuhmann et al., 2021), which is a subset of the web-crawled LAION-2B dataset (Schuhmann et al., 2022) used to train CLIP. In Fig. 2a, we plot the frequency of occurrence of "background" among other VOC class names. We observe that "background" is significantly more frequent than all other words, hinting that it is widely available not only in CLIP training data but also in web-crawled data in general.

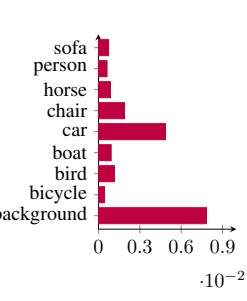
(a) Freq. of VOC concepts.

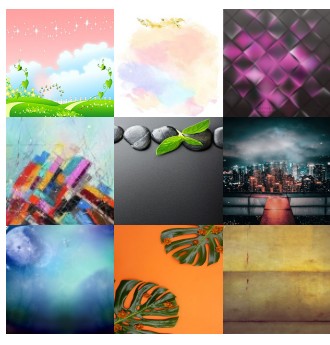
(b) "background" in caption

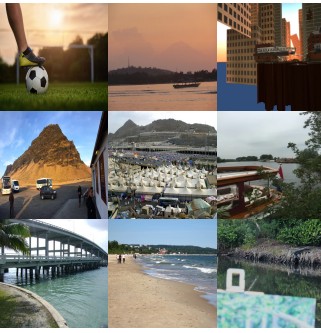
(c) "in the background" in caption

Figure 2: **Statistics about "background" in metadata of web-crawled datasets.** (a) Frequency of some of the concepts from VOC dataset in LAION-400M caption samples. Examples of images in web-crawled data with a caption including the words "background" (b) or "in the background" (c).

Fig. 2b shows images sampled from the LAION dataset that have a caption containing "background". We observe that they display a high diversity in colors and textures. Images captioned with "in the background" (Fig. 2c) appear to be more photo-oriented. We believe that the combination of a high frequency of the "background" word in the dataset and the diversity of associated images make it

a good generic contrastive concept, and hence make $\mathcal{CC}^{BG}$ a baseline. However, superior results have been obtained by applying well-designed tricks to handle the background (Wysoczańska et al., 2024a;b; Cha et al., 2023; Bousselham et al., 2024), emphasizing the necessity of applying something more than simply "background".

An option is to define a generic background class list, as done by CLIPpy (Ranasinghe et al., 2023) or CAT-Seg (Cho et al., 2024), which adds to the concept "background" a fixed list of concepts potentially appearing in the background, e.g., "sky", "forest", "building", to be discarded. First, since these visual concepts are intended to be discarded, it would not be possible to query them. Second, such a list is defined at the dataset level, making it domain-specific. As it is not possible to exhaustively describe all visual concepts appearing in any "background" (without prior knowledge of the domain or dataset), we propose to generate such complements specifically per query as discussed below.

### 3.3 AUTOMATIC CONTRASTIVE CONCEPTS ($\mathcal{CC}$) GENERATION

To generate contrastive concepts that are query-specific but also domain-agnostic, the only data we can then leverage are (i) the VLM's training data, or (ii) unspecific external data. As we focus on text-based contrasts, we can (i) exploit the large vocabulary of concepts used for VLM training, or (ii) generate prompts via an LLM. Finally, as we want good contrasts, we have to find hard negatives. These are concepts that surround queries in images. To gather them, we can (i) look for word co-occurrences in training data, or (ii) ask an LLM to list such concepts. Sec. 3.3.1 investigates option (i), and Sec. 3.3.2, option (ii).

### 3.3.1 MINING CO-OCCURRENCE-BASED CONTRASTIVE CONCEPTS ($\mathcal{CC}^D$)

As discussed above, ambiguity in segmentation for unsupervised approaches arises from co-occurrences in training data. Yet, OVSS does a better job when being prompted to create segments simultaneously for co-occurring concepts. To list contrastive concepts specific to a given query $q$, we thus propose to use information of *co-occurrence* in the VLM training captions. For efficiency, we construct *offline* a *co-occurrence dictionary*, built for a large lexicon of textual concepts extracted from the captions. We note $\mathcal{CC}^D_q$ the co-occurrence-based contrastive concepts we extract for a query $q$ based on this lexicon.

**Co-occurrence extraction.** We consider as lexicon a set of textual concepts $\mathcal{T}$ extracted from captions of the VLM training dataset and construct the co-occurrence matrix $X \in \mathbb{N}^{|\mathcal{T}| \times |\mathcal{T}|}$. Concretely, two concepts $\{i, j\} \subset \mathcal{T}$ co-occur if they appear simultaneously in the caption of an image. $X_{i,j}$ counts the number of times concepts $\{i, j\}$ co-occur in some images. Next, we normalize the symmetric matrix $X$ row-wise by the number of occurrences of concept $i$ in the dataset, producing the frequency matrix $\hat{X}$. We then consider only concepts with frequent co-occurrences: for each $i \in \mathcal{T}$, we select concepts $\mathcal{T}_i = \{j \in \mathcal{T} \mid \hat{X}_{i,j} > \gamma\}$, for some frequency threshold $\gamma$. Selecting only a few contrastive concepts in this way is also consistent with the fact that we target online segmentation: we need to be mindful of the computational costs.

**Concept filtering.** To improve the quality of selected contrastive concepts $\mathcal{T}_i$, we design a simple filtering pipeline. For each target concept $i \in \mathcal{T}$ (which can be thought of as a future query), we remove from $\mathcal{T}_i$ any concept that might interfere with $i$ and induce false negatives. First, we discard uninformative words that appear in captions: {"image", "photo", "picture", "view"}. Then, we remove *abstract* concepts, such as "liberty". To do so, we ask an LLM whether a given word can be visible or not in an image (more details in Appendix C.2). We also filter out concepts that are too semantically similar to target concept $i$, e.g., such that their cosine similarity with $\phi_T(i)$ is more than a threshold $\delta$. We also consider an alternative approach to filtering, which uses the structured ontology WordNet (Fellbaum, 1998) to remove the $\mathcal{CC}$s that possibly interfere with $q$. However, our experiments, which are discussed in Appendix B.2, show that our proposed filtering mechanisms based on dataset statistics are more effective.

**Generalization to arbitrary concepts.** So far, we discussed how to select contrastive concepts $\mathcal{CC}^D_i$ for a target concept $i \in \mathcal{T}$. Now when we are given an arbitrary textual query $q$, to make the

generation of contrastive concepts truly open-vocabulary, we first find in the CLIP space the nearest neighbor $i$ of $q$ in $\mathcal{T}$ and then use for $q$ the contrastive concepts of $i$: $\mathcal{CC}_q^D = \mathcal{CC}_i^D$.

### 3.3.2 PROMPTING AN LLM TO GENERATE CONTRASTIVE CONCEPTS $(\mathcal{CC}^L)$

Instead of extracting contrastive concepts from the VLM training set, we investigate here another strategy, which is generating them using an LLM. For a given text query $q$, we ask an LLM to directly generate contrastive concepts $\mathcal{CC}_q^L$, without the need for subsequent filtering. To that end, we design a prompt that excludes potential synonyms, meronyms (e.g., "wing" for "plane"), or possible contents (e.g., "wine" for "bottle"). We present a shorter version of the prompt in Fig. 3 and include the complete version in Appendix C.2.

*You are a helpful AI assistant with visual abilities. Given an input object **O**, I want you to generate a list of words related to objects that can be surrounding input object **O** in an image to help me perform semantic segmentation.*

Figure 3: **An abbreviated version of the prompt** we use to generate $\mathcal{CC}^L$.

Using an LLM has the benefit of producing specific contrastive concepts $\mathcal{CC}_q$ for any target query $q$, without returning to a fixed and practically limited lexicon.

## 4 EVALUATION

### 4.1 EVALUATING OPEN-WORLD SEGMENTATION

We discuss here our evaluation protocols and present our new metric IoU-single specifically designed to evaluate open-world segmentation.

**Evaluation datasets.** We conduct our experiments on six datasets widely used for the task of zero-shot semantic segmentation (Cha et al., 2023), fully-annotated COCO-Stuff (Caesar et al., 2018), Cityscapes (Cordts et al., 2016) and ADE20K (Zhou et al., 2019) and object-centric VOC (Everingham et al., 2012), COCO-Object (Caesar et al., 2018) and Context (Mottaghi et al., 2014), when considering "background" pixels. We treat the input images following the protocol of (Cha et al., 2023), which we detail in Appendix A.

**Our IoU-single metric.** To better evaluate the ability of a method to localize a visual concept when given *no other information*, we propose the IoU-single metric. It modifies the classic IoU by considering each concept independently and then averaging. Concretely, we individually segment each class annotated in the dataset for the considered image, thus with $|Q| = 1$. The IoU-single is then the average of each IoU with the corresponding ground-truth class segment. In Appendix A.3, we give a more intuitive illustration of our metric in Fig. 6 and a pseudo-code in Algorithm 1. If a dataset contains a *background* class, we do not consider it in the mIoU calculation.

**Classic mIoU evaluation.** We also evaluate the impact of using our $\mathcal{CC}$ in the classic mIoU scenario on the datasets that consider "background" as a class, i.e., VOC and COCO-Object. We prompt at once all dataset classes together with their $\mathcal{CC}$s, using our multiple-query strategy discussed in 3.1. We then assign pixels that fall into any of the $\mathcal{CC}$s to "background", ensuring that none of the concepts competes with the dataset queries. It allows us to verify if our $\mathcal{CC}$s can act as background without hurting the performance on foreground classes.

### 4.2 EVALUATED METHODS

**Test-time contrastive concepts.** For $\mathcal{CC}^D$ generation, we use the statistics gathered by Udandarao et al. (2024) for four thousand common concepts in the LAION-400M dataset, which is a subset of LAION-2B (Schuhmann et al., 2022) and which is used to train CLIP (Radford et al., 2021). We filter contrastive concepts using a low co-occurrence threshold $\gamma = 0.01$ and a high CLIP similarity threshold $\delta = 0.8$. In the classic mIoU scenario, we use a threshold $\beta = 0.9$ to account for possible similarities between one query and contrastive concepts close to the other queries. We discuss the selection of these values in Appendix B.1. To generate $\mathcal{CC}^L$, we use the recent Mixtral-8x7B-Instruct

model (Jiang et al., 2024). More details about the setup can be found in Appendix C.1 alongside our designed prompts in Appendix C.2. In our experiments, unless stated otherwise, we include "background" in all $\mathcal{CC}$'s: $\mathcal{CC}^D \leftarrow \{\text{"background"}\} \cup \mathcal{CC}^D$ and $\mathcal{CC}^L \leftarrow \{\text{"background"}\} \cup \mathcal{CC}^L$.

**Baselines.** To evaluate the impact of using contrastive concepts, we experiment on 5 popular or state-of-the-art methods, one of which (MaskCLIP) using 3 different backbones, thus resulting in 7 different segmenters, which we believe are representative of the current OVSS landscape. Concretely, we study two training-free methods that directly exploit the CLIP backbone, namely MaskCLIP (Zhou et al., 2022) and GEM (Bousselham et al., 2024), where MaskCLIP may exploit different OpenCLIP backbones (Ilharco et al., 2021) pre-trained either on LAION (Schuhmann et al., 2022), MetaCLIP (Xu et al., 2024), or by default on the original OpenAI training data (Radford et al., 2021). We also include TCL (Cha et al., 2023), CLIP-DINOiser (Wysoczańska et al., 2024b) and the supervised CAT-Seg (Cho et al., 2024). Details on the evaluation protocol, including background handling strategies, can be found in Appendix A. All compared methods use CLIP ViT-B/16.

### 4.3 CONTRASTIVE CONCEPTS GENERATION RESULTS

We first present in Tab. 1 results obtained with our IoU-single metric on 3 datasets, namely ADE20K, Cityscapes and VOC. We compare results when using different $\mathcal{CC}$'s proposed in this work. We also include results when having access to privileged information ($\mathcal{CC}^{PI}$), i.e., the list of concepts present in images as given by the evaluation dataset. More results can be found in Appendix Tab. 8.

| Method | CLIP training data | VOC | | | Cityscapes | | | | ADE20k | | | |
|---|---|---|---|---|---|---|---|---|---|---|---|---|
| | | $\mathcal{CC}^{BG}$ | $\mathcal{CC}^L$ | $\mathcal{CC}^D$ | $\mathcal{CC}^{BG}$ | $\mathcal{CC}^L$ | $\mathcal{CC}^D$ | $\mathcal{CC}^{PI}$ | $\mathcal{CC}^{BG}$ | $\mathcal{CC}^L$ | $\mathcal{CC}^D$ | $\mathcal{CC}^{PI}$ |
| MaskCLIP | OpenAI | 44.2 | 52.2 | **53.4** | 15.0 | **22.5** | 22.0 | 30.6 | 20.2 | 23.5 | **25.2** | 29.8 |
| DINOiser | LAION-2B | 59.3 | 63.1 | **64.7** | 23.2 | **30.6** | 27.3 | 36.0 | 28.9 | 29.7 | **31.6** | 35.5 |
| TCL | TCL's | 52.9* | 52.6* | **53.6*** | 9.8 | **26.3** | 22.0 | 29.7 | 14.9* | 25.9 | **26.5** | 32.6 |
| GEM | MetaCLIP | 48.6* | 61.3* | **64.6*** | 14.5* | **21.5** | 14.6 | 20.6 | 21.5* | 26.3 | **29.1** | 33.0 |
| CAT-Seg | OpenAI | 52.8 | **69.5** | 67.7 | – | – | – | – | 25.7 | 38.4 | **39.7** | 46.8 |

Table 1: **Benefits of $\mathcal{CC}$ measured in IoU-single.** '*' indicates that the method's original background handling is applied, if any and provided it gives the best results. Note that CAT-Seg input resolution is 640x640, whereas it is 448x448 for all the other methods. We note $\mathcal{CC}^{PI}$ the unrealistic setup where we have access to all of the dataset classes and use them as systematic contrastive concepts (except for VOC, as its annotations do not cover all pixels). Please note that $\mathcal{CC}^{BG}$ is our baseline.

**"Background" is not enough.** We start by analyzing the overall impact of our proposed $\mathcal{CC}$s. In all cases, we observe a significant improvement when using contrastive concepts $\mathcal{CC}^D$ and $\mathcal{CC}^L$ compared to the $\mathcal{CC}^{BG}$. Even for object-centric VOC where $\mathcal{CC}^{BG}$ already provides a strong baseline, our proposed $\mathcal{CC}$ generation methods bring significant gains ranging from 0.7 to 16.7 points. Interestingly, test-time $\mathcal{CC}$s also work well for supervised CAT-Seg, showing that our method is beneficial for open-vocabulary segmenters with all levels of supervision.

$\mathcal{CC}^L$ **generalize better to domain-specific datasets.** For both VOC and ADE20K, the co-occurrence-based $\mathcal{CC}^D$ outperforms most of the time the LLM-based $\mathcal{CC}^L$, with a margin ranging from 0.6 to 2.8 points. However, this trend does not hold for Cityscapes, where $\mathcal{CC}^L$ gives the best results for all methods. In particular, Cityscapes is a dataset of urban driving scenes that contains images depicting a few recurring concepts. This may suggest that LLMs can produce better results than $\mathcal{CC}^D$ for such domain-specific tasks. We also note that $\mathcal{CC}^L$ generally produces fewer $\mathcal{CC}$s, but we do not observe a correlation between segmentation performance and $|\mathcal{CC}|$, as shown in Appendix D.

**Test-time concepts are different from train-time concepts.** We also observe that $\mathcal{CC}^{PI}$ results overall do not exceed 50% mIoU. The segmentation quality might thus be limited by the VLM capacity or by a mismatch between the dataset classes and the training data. Well-designed prompt engineering could help address this issue (Roth et al., 2023) and improve segmentation results.

**Classic mIoU evaluation.** Additionally, in Tab. 2, we present results with the standard mIoU for MaskCLIP and CLIP-DINOiser (both with the LAION-2B backbone). We report results with various contrastive concepts ($\mathcal{CC}$) and the original background handling strategy when applicable. We observe that in all cases, the results with $\mathcal{CC}^D$ and $\mathcal{CC}^L$ are better than baseline $\mathcal{CC}^{BG}$. We also notice that for DINOiser the results are on par with the ones obtained with the saliency (noted 'sal.') originally proposed by Wysoczańska et al. (2024b). This shows that integrating our contrastive concepts does not hurt performance in the classic mIoU setup. We provide more results in Tab. 7.

| Method | | Bkg. | Object | VOC |
|---|---|---|---|---|
| MaskCLIP | $\mathcal{CC}^{BG}$ | 17.8 | 35.1 | |
| | $\mathcal{CC}^L$ | 25.9 | 46.2 | |
| | $\mathcal{CC}^D$ | 25.1 | 46.4 | |
| DINOiser | sal. | 34.8 | 62.1 | |
| | $\mathcal{CC}^{BG}$ | 29.5 | 54.0 | |
| | $\mathcal{CC}^L$ | 35.0 | 60.8 | |
| | $\mathcal{CC}^D$ | 33.3 | 60.4 | |

Table 2: **Results w/ mIoU.**

## 4.4 ABLATION STUDIES

| co-occ. | no abs. | sem. sim. | MaskCLIP | TCL | DINOiser |
|---|---|---|---|---|---|
| ✓ | | | 20.2 | 22.4 | 23.9 |
| ✓ | ✓ | | 20.9 | 23.2 | 25.5 |
| | ✓ | ✓ | 18.4 | 20.0 | 26.3 |
| ✓ | ✓ | ✓ | **25.2** | **26.0** | **31.6** |

| Method | Cityscapes w/o | w/ | ADE20k w/o | w/ |
|---|---|---|---|---|
| MaskCLIP | 22.3 | **22.5** | 22.5 | **23.5** |
| DINOiser | 30.3 | **30.6** | 27.5 | **29.7** |
| TCL | 26.0 | **26.2** | 25.4 | **26.3** |
| GEM | 21.3 | **21.4** | 25.7 | **26.1** |

| MaskCLIP w/ CLIP training set | VOC $\mathcal{CC}^{BG}$ | $\mathcal{CC}^L$ | $\mathcal{CC}^D$ |
|---|---|---|---|
| LAION-2B | 47.9 | 51.8 | **53.8** |
| OpenAI | 44.2 | 52.2 | **53.4** |
| MetaCLIP | 46.8 | **50.6** | 50.0 |

(a) **Impact of filtering in $\mathcal{CC}^D$ on ADE20K (%IoU-single).**

(b) **Adding "background" or not to our LLM-based $\mathcal{CC}^L$.**

(c) **Impact of pre-training dataset on VOC (%IoU-single).**

Table 3: **Ablation studies.** (a) The impact of filtering steps: 'co-occ.' is the co-occurrence-based filtering; 'no abs.' is the removal of abstract concepts; 'sem. sim.' is the semantic-similarity filtering. (b) Relevance of adding "background" to $\mathcal{CC}^L$. (c) Varying the pre-training dataset.

$\mathcal{CC}^D$ **concept filtering.** In Tab. 3a, we analyze the impact of the different filtering steps discussed in Sec. 3.3.1 on the challenging ADE20K dataset. We observe that each step boosts results by removing noisy or detrimental concepts. The largest gain is obtained when filtering highly similar ('sem. sim.') concepts. We also note that the improvement is consistent for all methods. We report the performance without the co-occurrence thresholding (w/o 'co-occ.') and observe a significant degradation. More experiments in Appendix B.2 suggest that ontology-based filtering (e.g., using WordNet) does not help and can even be harmful.

**Adding "background" to $\mathcal{CC}^L$.** In Tab. 3b, we study the influence of adding the word "background" to the set of contrastive concepts $\mathcal{CC}^L$ generated with the LLM. We observe that it is always beneficial, in most cases with little gain, except on ADE20k where the gain is up to 2.2 IoU-single pts.

**Impact of the pre-training dataset.** Tab. 3c shows the results of MaskCLIP with different datasets used to train CLIP. We observe that using $\mathcal{CC}^D$ always gives a boost over using "background" alone ($\mathcal{CC}^{BG}$) across all pre-training datasets, including on the highly-curated MetaCLIP. However, we notice that for MetaCLIP, $\mathcal{CC}^L$ gives even better results, suggesting that leveraging LLMs can also be more profitable with backbones pre-trained on carefully-curated datasets.

## 4.5 QUALITATIVE RESULTS

In Fig. 4, we present qualitative examples when using different contrastive concepts proposed in this work. We compare $\mathcal{CC}^L$ and $\mathcal{CC}^D$ with ground truth (GT) and baseline $\mathcal{CC}^{BG}$. For both $\mathcal{CC}^L$ and $\mathcal{CC}^D$, we present the output segmentation mask for the queried concept together with its contrastive concepts (noted *all*) as well as the single queried concept (noted *single*), where $\mathcal{CC}$s are discarded. We observe that the output masks produced by our methods are more accurate, removing the noise from related concepts, e.g. "tree" for the bird, or "sofa" for the "bed".

**Generalization to arbitrary concepts.** Fig. 5 presents results when prompting queries that are not included in the subset of concepts $\mathcal{T}$ extracted from the VLM training dataset, such as "muffin" or "cavalier" (a dog breed). We show the closest neighbor for the query $q$ below each example and

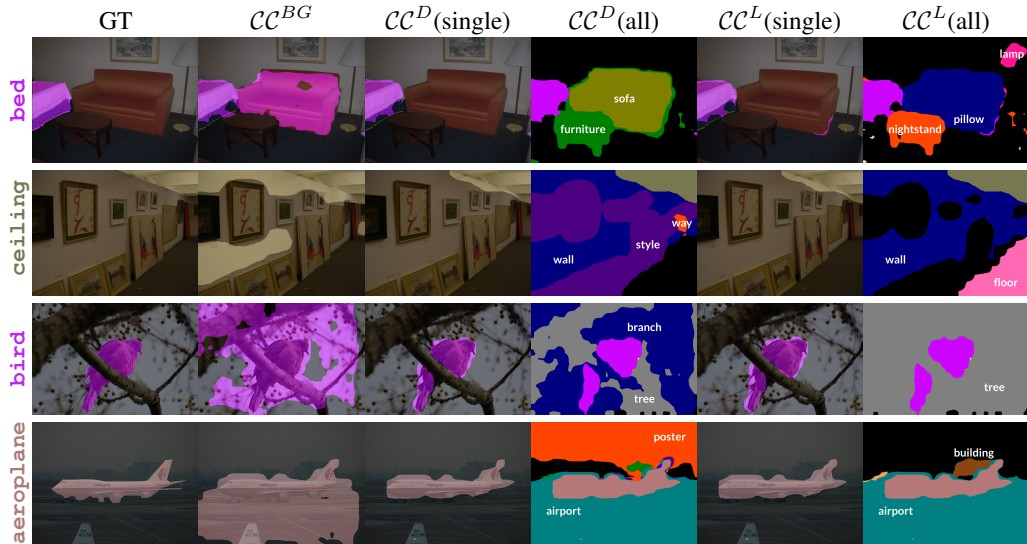

Figure 4: **Qualitative results.** We show segmentation examples from ADE20K (1st and 2nd row) and Context (3rd and 4th row), with segments produced by CLIP-DINOiser. For $\mathcal{CC}^D$ and $\mathcal{CC}^L$, we additionally show the joint segmentation of all contrastive classes (all).

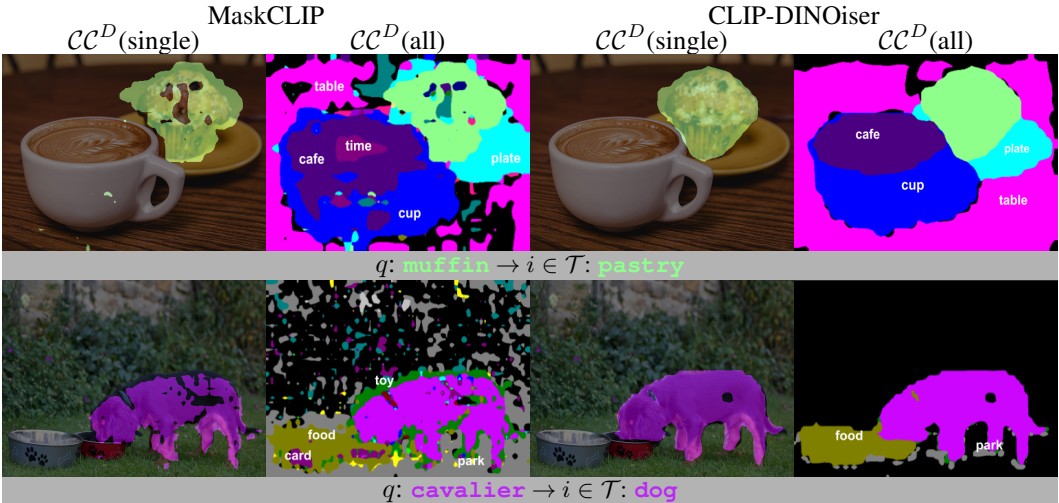

Figure 5: **In the wild examples**. We visualize results for MaskCLIP and CLIP-DINOiser for query concepts beyond $\mathcal{T}$. The closest neighbour to a query is presented below each example (grey row).

visualize masks for both MaskCLIP and CLIP-DINOiser. We observe that the $\mathcal{CC}^D$ generation method leveraging statistics from pre-training datasets is also robust to examples outside of the co-occurrence dictionary by accurately mapping $q$ to its closest concept in $\mathcal{T}$, e.g., mapping "cavalier" to "dog".

## 5 CONCLUSION

In this work, we identify limitations of the current evaluation setup for open-vocabulary semantic segmentation tasks, which are inherited from close-world evaluation benchmarks. To bridge the gap between close- and open-world setups, we propose the single-class segmentation scenario. We study the limitations of current state-of-the-art models when we assume no prior access to in-domain classes and propose to automatically discover contrastive concepts $\mathcal{CC}$ that are useful to better localize any queried concept. To do so, we propose two methods leveraging either the distribution of co-occurrences in the VLM's training set or an LLM to generate such $\mathcal{CC}$. Our results show the generalizability of our proposed method across several setups.

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

APPENDIX

In this appendix,

- we start by providing details on the evaluation in Sec. A: evaluation protocol (Sec. A.1), approaches to the background handling of the considered baselines (Sec. A.2), and details of the IoU-Single metric (Sec. A.3).

- Next, in Sec. B, we present more quantitative results and studies, including hyperparameter selection (Sec. B.1), filtering based on WordNet ontology (Sec. B.2), classic mIoU results (Sec. B.3), and further quantitative (Sec. B.3) and qualitative (Sec. B.4) results. We also discuss limitations (Sec. 5).

- In Sec. C, we provide details about LLM and the used prompts, together with examples of LLM-generated contrastive concepts.

- Sec. D presents performance vs. the number of contrastive concepts when considering $\mathcal{CC}^D$ and $\mathcal{CC}^L$.

## A DETAILS ON THE EVALUATION

### A.1 EVALUATION PROTOCOL

In our experiments, we follow the evaluation protocol of Cha et al. (2023). We use MMSegmentation implementation (Contributors, 2020) with a sliding window strategy and resize input images to have a shorter side of 448. In the case of CAT-Seg, we retain the original model framework and integrate IoU-single into Detectron (Wu et al., 2019). We also use its evaluation protocol, meaning that the input images differ from other evaluated methods, i.e., with an input image size of 640x640. Regarding the text prompts, we keep the native prompting of each method to stay as close as possible to the methods.

### A.2 BACKGROUND HANDLING OF BASELINES

We detail here the different strategies employed in the methods that we evaluate to handle the background.

**TCL** (Cha et al., 2023) applies thresholding and considers pixels with maximal logit $\leq 0.5$ to be in the background, where the logits are the cosine similarities of the visual embedding with the embedding of queries.

**GEM** (Bousselham et al., 2024) applies a background handling strategy only for Pascal VOC. It only predicts the foreground classes. The background is obtained by thresholding the softmax-normalized similarity between the patch tokens and the text embedding of each class name. The threshold is fixed (set to 0.85). In our experiments with VOC, we explore the performance of GEM both with and without background handling and report each time the better score. For other datasets than VOC, we apply only our methods.

**MaskCLIP** (Zhou et al., 2022) does not use any dedicated mechanism for background. Therefore, we do not report the original setup for it.

**CLIP-DINOiser** (Wysoczańska et al., 2024b) leverages a foreground/background saliency strategy which focuses on foreground pixels. In that case, the foreground/background is defined following FOUND (Siméoni et al., 2023), which focuses on objectness and mainly discards pixels corresponding to stuff-like classes, which might also be of interest.

**CAT-Seg** (Cho et al., 2024) does not apply any background handling strategy. Instead, for VOC they create a list of potential background classes and use them as "dummy" classes. This approach is the closest to what we propose in our work. In practice, for the VOC dataset, the authors use class names from the Context dataset, an extension of VOC with +40 class names.

---

**Algorithm 1:** IoU-single

---

**input** : $I$ – input image: $I \in \mathbb{R}^{H \times W \times 3}$
$\quad\quad\quad$ $Y$ – ground-truth annotations of $I$: $gt \in \mathbb{N}^{H \times W \times 1}$
$\quad\quad\quad$ $T$ – ground-truth text labels
$\quad\quad\quad$ $CC$ – a dictionary of contrastive concepts per query
$\quad\quad\quad$ `model` – segmenter producing pixel-level predictions given text queries
**output :** mean IoU-single, a mIoU score for a single-query scenario for a given image
**procedure** `IoUsingle`$(I, Y)$:
$\quad$ *// Get unique classes from $Y$*
$\quad$ $gt_{cls} \leftarrow$ `unique`$(Y)$
$\quad$ scores $\leftarrow \emptyset$
$\quad$ **for** $i \in gt_{cls}$ **do**
$\quad\quad$ $q \leftarrow T_i$
$\quad\quad$ *// Text prompts include query $q$ and contrastive concepts of $q$*
$\quad\quad$ $t_q \leftarrow q \cup CC_q$
$\quad\quad$ *// Get model predictions for given prompt set*
$\quad\quad$ $\hat{y} \leftarrow$ `model`$(I, t_q)$
$\quad\quad$ *// Get binarized version of predicted mask*
$\quad\quad$ $\hat{y} \leftarrow$ `binarize`$(\hat{y}, i)$
$\quad\quad$ *// Get ground-truth binary mask for gt class $i$*
$\quad\quad$ $y \leftarrow$ `binarize`$(Y, i)$
$\quad\quad$ *// Record corresponding IoU*
$\quad\quad$ scores $\leftarrow$ scores $\cup$ `IoU`$(\hat{y}, y)$
$\quad$ **end for**
**return** `mean`(scores)

---

### A.3 ABOUT THE IoU-SINGLE METRIC

We present in Fig. 6 an illustration of our proposed metric IoU-single. We illustrate the difference between the standard mIoU metric (dataset-driven mIoU), where all the concepts present on an image are considered at once. On the contrary, our IoU-single considered each of the present concepts separately to measure the single-class segmentation ability of open vocabulary semantic segmentation methods.

We also present in Algorithm 1 a pseudo-code of our metric. (The actual code of this metric, along with the code used in our experiments, will be available upon publication.)

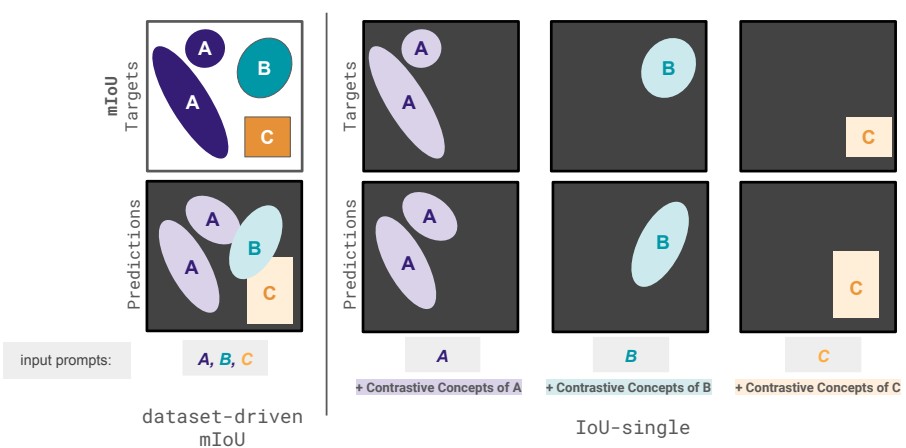

Figure 6: **Illustration of our IoU-single metric**.

| Method | CLIP tr. data | values of $\gamma$ | | | | | values of $\delta$ | | | | |
|---|---|---|---|---|---|---|---|---|---|---|---|
| | | 0.001 | 0.005 | 0.01 | 0.015 | 0.02 | 0.95 | 0.9 | 0.85 | 0.8 | 0.75 |
| MaskCLIP | OpenAI | 24.4 | 26.0 | 24.8 | 24.4 | 23.2 | 19.9 | 21.0 | 23.0 | 24.4 | 22.8 |
| | Laion2B | 25.8 | 27.8 | 27.4 | 26.0 | 25.4 | 23.0 | 24.1 | 26.4 | 27.4 | 24.6 |
| | MetaCLIP | 22.0 | 24.1 | 24.4 | 23.8 | 23.4 | 22.7 | 23.7 | 25.9 | 27.2 | 23.7 |
| DINOiser | Laion2B | 24.4 | 27.2 | 27.9 | 27.9 | 27.7 | 23.5 | 24.6 | 26.4 | 27.9 | 26.9 |

Table 4: **Parameter study of $\gamma$ and $\delta$.** Selection (marked in grey) of the hyperparameters $\gamma$ and $\delta$ with IoU-single on 100 randomly-selected images in ADE20k training dataset.

| Method | CLIP training data | 1.0 | 0.95 | 0.9 | 0.85 | 0.8 |
|---|---|---|---|---|---|---|
| MaskCLIP | OpenAI | 26.0 | 40.4 | 41.1 | 39.1 | 32.1 |
| | Laion2B | 35.3 | 43.7 | 44.0 | 44.6 | 42.2 |
| | MetaCLIP | 24.4 | 39.1 | 40.3 | 34.3 | 30.6 |
| DINOiser | Laion2B | 51.3 | 57.8 | 58.6 | 58.8 | 55.2 |
| TCL | TCL's | 37.2 | 47.6 | 47.7 | 47.1 | 47.7 |

Table 5: **Selection of $\beta$ with classic mIoU** on 100 randomly-selected images in the VOC training dataset. Results are reported for $\mathcal{CC}^L$.

| Method | MaskCLIP | TCL | DINOiser |
|---|---|---|---|
| $\mathcal{CC}^D$ | **25.2** | 26.0 | **31.6** |
| $\mathcal{CC}^D$ + WordNet | **25.2** | **26.4** | 26.3 |
| $\mathcal{CC}^D$ + WordNet − sem. sim. | 21.0 | 23.4 | 25.8 |

Table 6: **Ontology-based (WordNet) filtering** out synonyms, meronyms, hyponyms and hypernyms (at depth 1) from $\mathcal{CC}^D$. Results are reported on ADE20K, as %IoU-single.

# B MORE QUANTITATIVE RESULTS

## B.1 HYPERPARAMETER SELECTION

In this section, we discuss the selection of hyperparameters for our $\mathcal{CC}$ generation. For the frequency threshold $\gamma$ and the cosine similarity threshold $\delta$, we randomly select 100 images from the training set of the ADE20K dataset and report IoU-single on this subset — which we observed was enough to select the values. We report in Tab. 4 a parameter study of both hyperparameters and mark in grey selected values, i.e., $\gamma = 0.01$ and $\delta = 0.8$. For $\gamma$, we observe that values $\gamma < 0.005$ are too low, most likely introducing too much noise in selected contrastive concepts.

Tab. 5 presents a parameter study of the cosine similarity of text queries $\beta$ in multi-query segmentation. Here, we randomly select 100 images from the VOC training set and report classic mIoU for different $\beta$ values. We select $\beta = 0.9$ because it gives the best result for most methods. We also note that controlling the similarity between query concepts and contrastive concepts in the multiple-query scenario is necessary. Not including this step (see results for $\beta = 1.0$) greatly degrades performance.

## B.2 ONTOLOGY-BASED FILTERING WITH WORDNET

Here, we discuss our experiments when using the WordNet ontology (Fellbaum, 1998) for $\mathcal{CC}^D$ filtering. Specifically, for each query concept, we extract synonyms and meronyms, as well as hyponyms and hypernyms in-depth 1 in the WordNet ontology. From the results in Tab. 6, we observe that adding such filtering on top of our *semantic similarity* filtering brings little to no improvement, suggesting that *semantic filtering* removes most of the contrastive concepts that interfere with a query

| Methods | Background handling | Type of $\mathcal{CC}$ | CLIP backbone | Training dataset | Dataset Context | Object | VOC |
|---|---|---|---|---|---|---|---|
| GroupViT | threshold | ∅ | scratch | CC12M+RedCaps | 18.7 | 27.5 | 50.4 |
| CLIP-DIY | saliency | ∅ | LAION-2B | - | 19.7 | 31.0 | 59.9 |
| TCL | threshold | ∅ | OpenAI | CC12M+CC3M | 24.3 | 30.4 | 51.2 |
| MaskCLIP† | ∅ | ∅ | OpenAI | - | 21.1 | 15.5 | 29.3 |
| MaskCLIP* | ∅ | ∅ | LAION-2B | - | 22.9 | 16.4 | 32.9 |
| MaskCLIP* (+$keys$) | ∅ | ∅ | LAION-2B | - | 24.0 | 21.6 | 41.3 |
| CLIP-DINOiser | ∅ | ∅ | LAION-2B | ImageNet (1k im.) | 32.4 | 29.9 | 53.7 |
| GEM | ∅ | ∅ | MetaCLIP | - | - | - | 46.8 |
| CLIP-DINOiser | saliency | ∅ | LAION-2B | ImageNet (1k im.) | – | 34.8 | **62.1** |
| | $\mathcal{CC}$ | $\mathcal{CC}^{BG}$ | LAION-2B | ImageNet (1k im.) | **32.4** | 29.5 | 54.0 |
| | $\mathcal{CC}$ | $\mathcal{CC}^{L}$ | LAION-2B | ImageNet (1k im.) | 31.3 | **35.0** | 60.8 |
| | $\mathcal{CC}$ | $\mathcal{CC}^{D}$ | LAION-2B | ImageNet (1k im.) | 31.8 | 33.3 | 60.4 |
| MaskCLIP | $\mathcal{CC}$ | $\mathcal{CC}^{BG}$ | LAION-2B | - | 23.6 | 17.8 | 35.1 |
| | $\mathcal{CC}$ | $\mathcal{CC}^{L}$ | LAION-2B | - | **22.5** | **25.9** | 46.2 |
| | $\mathcal{CC}$ | $\mathcal{CC}^{D}$ | LAION-2B | - | 23.2 | 25.1 | **46.4** |
| GEM | threshold | ∅ | MetaCLIP | - | **33.4*** | 27.4* | 46.6* |
| GEM | $\mathcal{CC}$ | $\mathcal{CC}^{L}$ | MetaCLIP | - | 31.6 | **35.7** | 60.0 |
| GEM | $\mathcal{CC}$ | $\mathcal{CC}^{D}$ | MetaCLIP | - | 32.1 | 35.5 | **60.5** |

Table 7: **Results with standard mIoU metric** when employing different contrastive concept generation strategies. '*' denotes our implementation, '†' denotes results from TCL (Cha et al., 2023), and 'MaskCLIP (+keys)' denotes keys refinement proposed in the original paper (Zhou et al., 2022). Training datasets include CC12M (Changpinyo et al., 2021), RedCaps (Desai et al., 2021), ImageNet (Deng et al., 2009), CC3M (Sharma et al., 2018).

concept. Furthermore, replacing *semantic similarity* with WordNet-based filtering yields significantly worse results than our proposed $\mathcal{CC}^{D}$.

### B.3 MORE QUANTITATIVE RESULTS

**State-of-the-art results under classic mIoU** In Tab. 7, we report the results under the classic mIoU metric for selected state-of-the-art methods on open-vocabulary semantic segmentation. For each of the methods, we detail the specific background handling techniques (if any), the CLIP backbone used as well as additional datasets used for training.

We notice that extending the dataset vocabulary with our generated contrastive concepts does not hurt the overall performance under a normal setup when all dataset labels are considered as prompts. For GEM and MaskCLIP we observe significant improvements over their original setups on VOC. This holds for both contrastive concept generation methods $\mathcal{CC}^{D}$ and $\mathcal{CC}^{L}$. Looking at the results of CLIP-DINOiser, we observe that saliency is still more effective in the object-centric scenario.

**More open-world evaluation results** Tab. 8 extends Tab. 1 and completes the results, obtained with the IoU-single metric on all the datasets that we considered.

### B.4 MORE QUALITATIVE RESULTS

More qualitative results are provided in Fig. 7, comparing $\mathcal{CC}^{D}$ to $\mathcal{CC}^{L}$.

| Method | CLIP dataset | Original | $CC^{PI}$ | $CC^{BG}$ | $CC^L$ | $CC^D$ |
|---|---|---|---|---|---|---|
| **VOC** | | | | | | |
| | LAION-2B | – | 49.9 | 47.9 | 51.8 | **53.6** |
| MaskCLIP | OpenAI | – | 47.1 | 44.2 | 52.2 | **53.4** |
| | MetaCLIP | – | 47.9 | 46.6 | **50.6** | 50.1 |
| CLIP-DINOiser | LAION-2B | 63.8* | 61.0 | 59.3 | 63.1 | **64.7** |
| TCL | TCL's | 52.9* | 53.0* | 52.9* | 52.6* | 53.6* |
| GEM | MetaCLIP | – | – | 48.6* | 61.3* | 64.6* |
| CAT-Seg | OpenAI | – | – | 52.8 | **69.5** | 67.7 |
| **Cityscapes** | | | | | | |
| | LAION-2B | – | 32.2 | 16.2 | **27.2** | 24.0 |
| MaskCLIP | OpenAI | – | 30.6 | 15.0 | **22.5** | 22.0 |
| | MetaCLIP | – | 30.0 | 13.6 | **24.6** | 23.3 |
| CLIP-DINOiser | LAION-2B | 20.8 | 36.0 | 23.2 | **30.6** | 27.3 |
| TCL | TCL's | 18.6* | 29.7 | 9.8 | **26.3** | 22.0 |
| GEM | MetaCLIP | – | 20.6 | 14.5* | **21.5** | 14.6 |
| **COCO-Stuff** | | | | | | |
| | LAION-2B | – | 34.1 | 26.4 | 28.8 | **29.5** |
| MaskCLIP | OpenAI | – | 33.6 | 24.1 | 28.4 | **28.8** |
| | MetaCLIP | – | 34.0 | 25.8 | **28.1** | 28.1 |
| CLIP-DINOiser | LAION-2B | 28.0* | 35.3 | 32.4 | 33.9 | **34.4** |
| TCL | TCL's | 25.0* | 34.7 | 17.4 | 29.5 | **30.6** |
| GEM | MetaCLIP | – | 38.3 | 22.9* | 32.2 | **33.6** |
| **ADE20k** | | | | | | |
| | LAION-2B | – | 33.2 | 22.7 | 26.8 | **27.8** |
| MaskCLIP | OpenAI | – | 29.8 | 20.2 | 23.5 | **25.2** |
| | MetaCLIP | – | 32.1 | 21.5 | 24.7 | **26.0** |
| CLIP-DINOiser | LAION-2B | 28.8* | 35.3 | 28.9 | 29.7 | **31.6** |
| TCL | TCL's | 14.8* | 32.6 | 14.9* | 25.9 | **26.5** |
| GEM | MetaCLIP | – | 33.0 | 21.5* | 26.3 | **29.1** |
| CAT-Seg | OpenAI | – | 46.8 | 25.7 | 38.4 | **39.7** |
| **COCO-Object** | | | | | | |
| | LAION-2B | – | 32.1 | 27.7 | **33.7** | 32.9 |
| MaskCLIP | OpenAI | – | 31.3 | 24.3 | **34.5** | 33.3 |
| | MetaCLIP | – | 30.9 | 27.4 | **32.2** | 31.1 |
| CLIP-DINOiser | LAION-2B | 38.8* | 38.9 | 35.5 | **41.6** | 39.9 |
| TCL | TCL's | 37.1* | 38.1 | 37.2* | **38.1*** | 37.2* |
| GEM | MetaCLIP | – | – | 31.4 | 39.7 | **40.1** |
| **Pascal Context** | | | | | | |
| | LAION-2B | – | 40.5 | 34.4 | 35.2 | **37.4** |
| MaskCLIP | OpenAI | – | 41.1 | 32.9 | 34.7 | **36.8** |
| | MetaCLIP | – | 41.1 | 32.6 | 34.2 | **35.8** |
| CLIP-DINOiser | LAION-2B | 33.9* | 45.8 | 41.5 | 41.6 | **44.2** |
| TCL | TCL's | 29.7* | 41.7 | 29.7* | 36.8 | **38.2** |
| GEM | MetaCLIP | – | – | 26.9 | 40.1 | **42.1** |

Table 8: Results on all datasets with our IoU-single metric defined in Sec. 4.1. '*' denotes the result when the original background handling gives the best results.

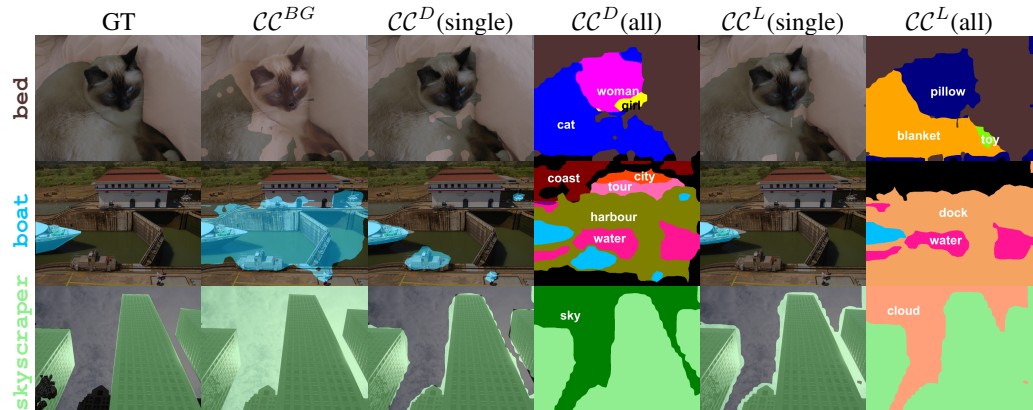

Figure 7: **More qualitative results** of CLIP-DINOiser with different $\mathcal{CC}$. Here we focus on cases where $\mathcal{CC}^D$ and $\mathcal{CC}^L$ give different results. For "boat" (2nd row), $\mathcal{CC}^L$ gives a better result providing a good $\mathcal{CC}$ ("dock"). On the other hand, for "skyscraper" (3rd row), $\mathcal{CC}^D$ yields slightly better results suggesting "sky" and not "cloud". Note that in this last example, $\mathcal{CC}^{BG}$ completely fails, possibly due to a difficult (uncommon) angle of view.

## B.5 LIMITATIONS

In many images, the objects of interest are surrounded by unrelated other objects or stuff. Contrastive concepts are then of little help, and the objects of interest are reasonably well segmented by only contrasting with "background". This is, in particular, the case for object-centric datasets such as VOC. Differently, in some images, the objects of interest are surrounded by other objects or stuff that are frequently co-occurring in the training set. Using specific contrastive objects helps the segmenter to better separate the objects of interest from the rest. Consequently, the average gain of using $\mathcal{CC}$ is good but possibly moderate. However, in some images and queries, the specific gain can be particularly high.

## C PROMPTING THE LLM

In this section, we provide more details about the LLM and the prompts used.

### C.1 THE LLM MODEL

We use the recent Mixtral-8x7B-Instruct model (Jiang et al., 2024), a sparse mixture of experts model (SMoE), finetuned for instruction following and released by Mistral AI. More precisely, we rely on the v0.1 version of its open weights available via the Hugging Face transformers library. We run the LLM in 4-bit precision with flash attention to speedup inference.

### C.2 THE PROMPTS USED FOR CONTRASTIVE CONCEPTS

We provide in Fig. 8 the prompt used to generate the contrastive concepts $\mathcal{CC}^L$ and in Fig. 9 the prompt used to predict whether a concept can be seen in an image or not in order to filter $\mathcal{CC}^D$.

In these prompts, we indicate the inserted input text as $\{q\}$. We follow Mixtral-8x7B Instruct's prompt template. In particular, we use  as the beginning of the string (BOS) special token, as well as *[INST]* and *[/INST]* as string markers to be set around the instructions.

For the generation of $\mathcal{CC}^L$, we also integrate a light post-processing step, ensuring that all generated lists have a unified format with coma separation. We do not apply any filtering or cleaning step to the LLM-generated results.

 [INST] You are a helpful AI assistant with visual abilities.

Given an input object O, I want you to generate a list of words related to objects that can be surrounding input object O in an image to help me perform semantic segmentation.

For example:

* If the input object is 'fork', you can generate a list of words such as '["bottle", "knife", "table", "napkin", "bread"]'.

* If the input object is 'child', you can generate a list of words such as '["toy", "drawing", "bed", "room", "playground"]'.

You should not generate synonyms of input object O, nor parts of input object O.

Generate a list of objects surrounding the input object $\{q\}$ without any synonym nor parts, nor content of it. Answer with a list of words. No explanation.

Answer: [/INST]

Figure 8: Prompt for $CC^L$ contrastive concept generation.

 [INST] Please specify whether $\{q\}$ is something that one can see.

Reply with 'yes' or 'no' only. No explanation.

Answer: [/INST]

Figure 9: Prompt for $CC^L$ visibility prediction.

 [INST] You are a helpful AI assistant with visual abilities.

Given an input object O, I want you to generate a list of words that are parts of an object O.

For example:

* If the input object is 'rabbit', you can generate a list of words such as '["paw", "tail", "fur", "ears", "muzzle"]'.

* If the input object is 'building', you can generate a list of words such as '["door", "window", "wall", "hall", "floor"]'.

Generate a list of parts of the input object $\{q\}$. Answer with a list of words. Do not give any word that is not a part of the input object. No explanation.

Answer: [/INST]

Figure 10: Prompt for part prediction.

## C.3 PART REMOVAL VIA LLM-PROMPTING

We also explore the possibility of removing suggested contrastive concepts that can be *parts* of query concepts. Note that in $CC^L$, we explicitly do it in the prompt itself (Fig. 10). Fig. 11 presents one of such examples when removing "wheel" from the $CC^D$ of query "bicycle" gives a slight improvement for MaskCLIP segmentation. However, we do not notice a particular improvement in the case of other segmentation methods since, typically, they refine the masks or feature maps to include localization priors. For example, in Fig. 11, the second row presents the same example for CLIP-DINOiser (DINOiser), where the improvement is marginal. Finally, we observe little or no quantitative improvement when applying part removal filtering on entire datasets. Therefore, we do not include it in our final method.

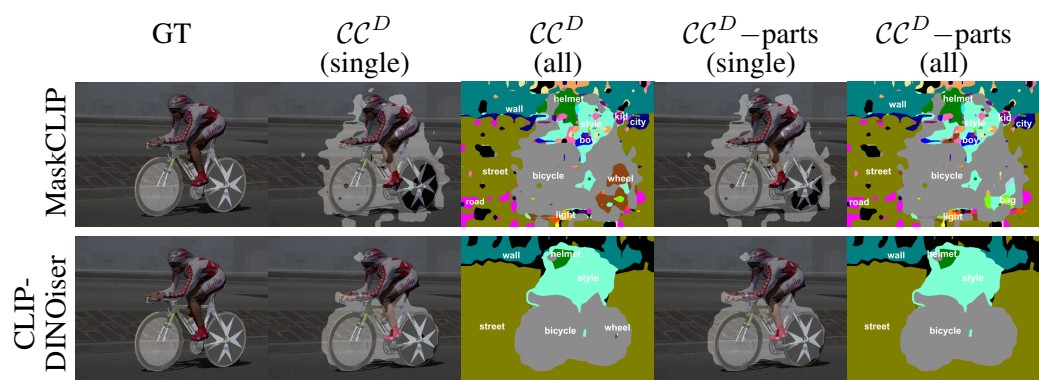

Figure 11: **Part removal.** We consider an example from Pascal Context with $q = \texttt{bicycle}$. We show the segmentation masks produced by MaskCLIP and CLIP-DINOiser for $\mathcal{CC}^D$, as well as for $\mathcal{CC}^D$ when parts of objects are removed ($\mathcal{CC}^D-$parts).

| Query $q$ | $\mathcal{CC}^L_q$ |
|---|---|
| road | building, tree, car, pedestrian, sky, streetlight, sidewalk, bicycle, parked car, traffic sign |
| sidewalk | building, street, car, tree, people, bike, road, park, sky, lane |
| building | sky, tree, road, car, park, people, lane, fence, house, field |
| wall | door, window, floor, ceiling, painting, light, chair, table, carpet, curtain |
| fence | grass, tree, house, car, path, post, gate, field, flowers, animals |
| pole | building, wire, tree, street, sky, fence, cable, road, banner, light |
| traffic light | road, car, building, pedestrian, sky, streetlight, traffic sign, parking meter |
| traffic sign | road, street, pole, vehicle, building, sky, pedestrian, curb, lane, light |
| vegetation | soil, tree, grass, water, animal, fence, field, sky, rock, sun |
| terrain | tree, sky, building, road, mountain, river, field, fence, vehicle, person |
| sky | tree, building, cloud, sun, bird, airplane, mountain, sea, sunset, cityscape |
| person | bike, road, car, tree, building, park, cityscape, nature, animal, sports equipment |
| rider | bicycle, road, nature, park |
| car | road, tree, building, person, parking |
| truck | road, car, building, tree, parking |
| bus | road, tree, building, sky, person, car, traffic light, bicycle, parking meter, street sign |
| train | track, grass, sky, building, platform, tree, sign, person, car, road |
| motorcycle | road, person, bike, car, traffic, building, nature, parking, city, scenery |
| bicycle | road, tree, person, park, building, grass, basket, helmet, traffic, path |

Table 9: Example of LLM-generated $\mathcal{CC}^L$ for Cityscapes.

## C.4 EXAMPLE OF GENERATED $\mathcal{CC}^L$

## D AVERAGE NUMBER OF CONTRASTIVE CONCEPTS VS PERFORMANCE

We present in Fig. 13 a scatterplot of performance vs the number of contrastive concepts when considering $\mathcal{CC}^D$ (Fig. 13(a)) and $\mathcal{CC}^L$ (Fig. 13(b)). The points correspond to the IoU-single scores per class obtained with CLIP-DINOiser on all datasets we evaluate. We observe no strong correlation between the number of contrastive concepts and performance, although there is a small mode of around 20 concepts when using $\mathcal{CC}^D$. We also observe that, on average, $|\mathcal{CC}^D| > |\mathcal{CC}^L|$.

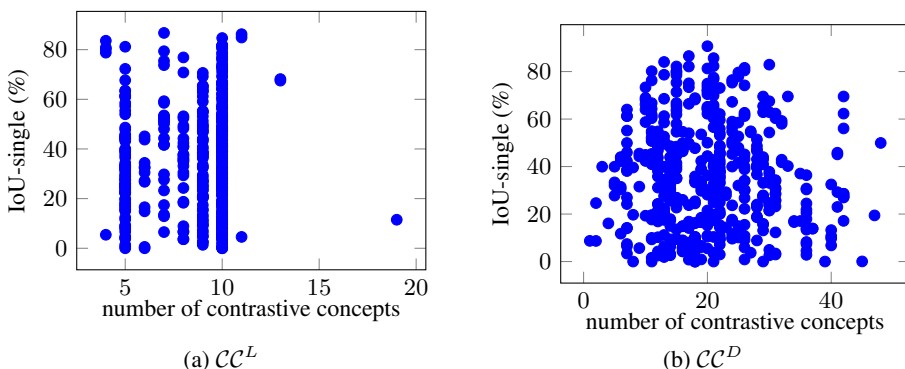

(a) $\mathcal{CC}^L$ (b) $\mathcal{CC}^D$

Figure 12: **Number of $\mathcal{CC}$ vs performance.** We compare the number of $\mathcal{CC}$ against the performance of CLIP-DINOiser for each class used in our evaluations (considering all datasets). Performance is reported with per class IoU-single %.

# E ADDITIONAL VISUALS

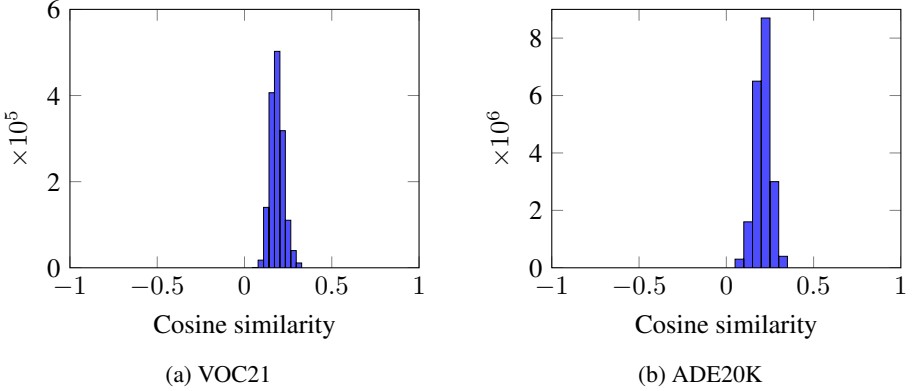

(a) VOC21 (b) ADE20K

Figure 13: **Distribution of maximum patch similarities with text prompts.** We plot histograms for 100 images of VOC21 (a) and ADE20K (b) of patch similarities in MaskCLIP.

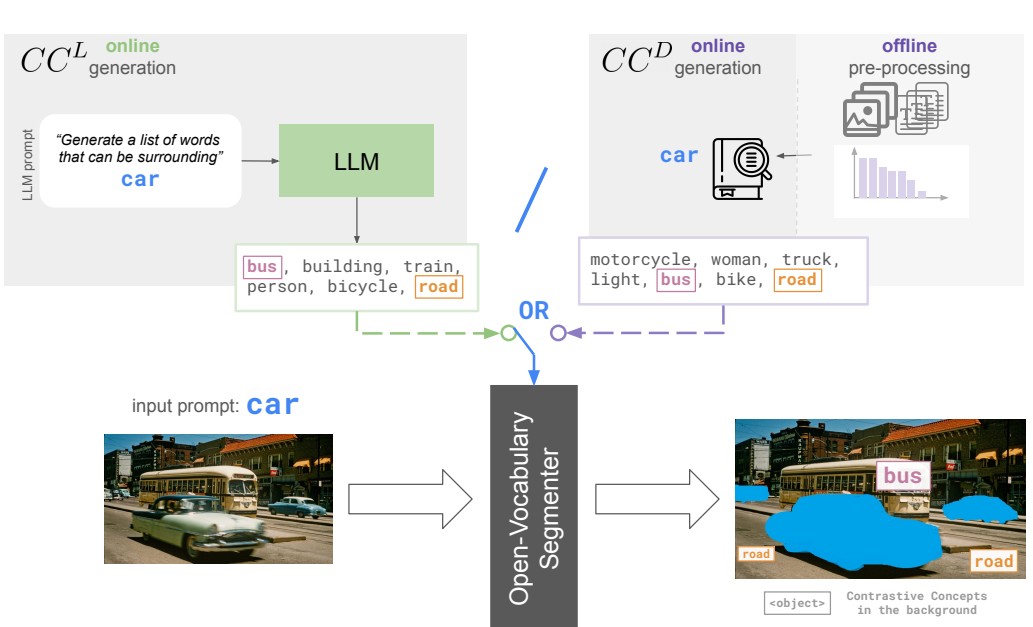

Figure 14: **Overview of our method.**

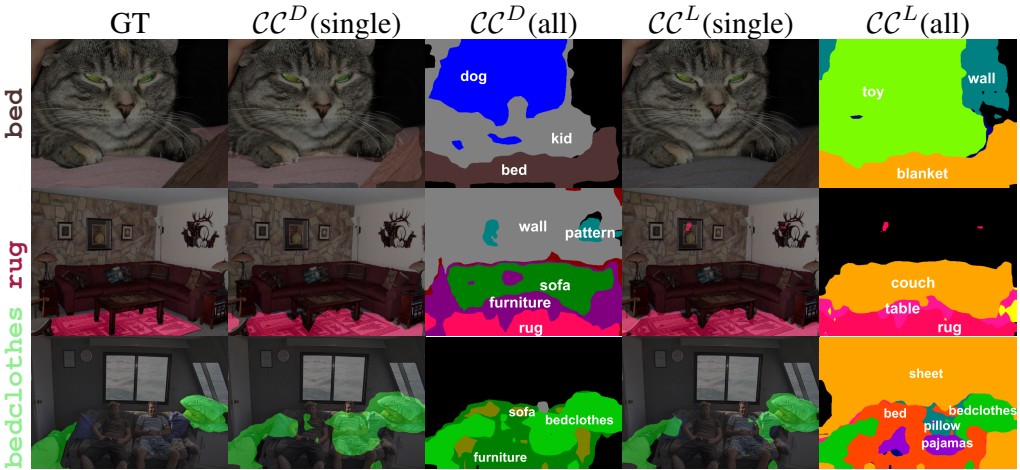

Figure 15: **Failure cases of our method.** We show examples of CLIP-DINOiser when one of the methods fails to generate accurate $\mathcal{CC}$. In the first example $\mathcal{CC}^L$ suggests "blanket" for "bed" which typically covers the query concept. In the second row, both methods fail to provide "floor" to contrast with "rug". Finally, in the third example, both methods fail to generate "person" to contrast with "bedclothes", however, $\mathcal{CC}^L$ suggest "pyjamas", which results in a better segmentation.

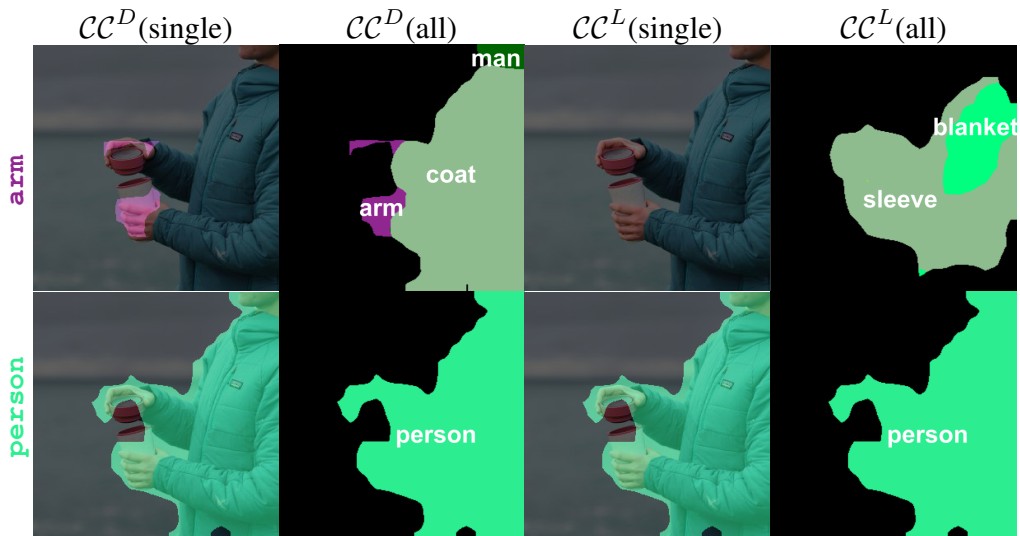

Figure 16: **Parts of objects handling.** We present here qualitative results with the method CLIP-DINOiser (Wysoczańska et al., 2024b) when prompting the prompts 'arm' and 'person' as suggested by the reviewer. We present results with both $\mathcal{CC}^D$ and $\mathcal{CC}^L$. We observe that the prompt 'person' is always well segmented when the part 'arm' suffers from occlusion with our $\mathcal{CC}$.

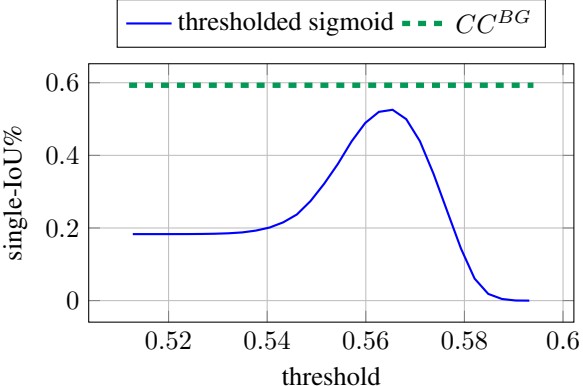

Figure 17: **Sigmoid experiments.** We replace softmax with sigmoid applied on individual patch-to-query prompt similarities. We show the variation of single-IoU% wrt. the threshold that is applied after sigmoid to decide on a positive vs. "background" class. To get the thresholds, we find the minimum and maximum values of the features after sigmoid and linearly sample 30 values in this range. We can see that the result is sensitive to the threshold value and does not reach the baseline of $CC^{BG}$.

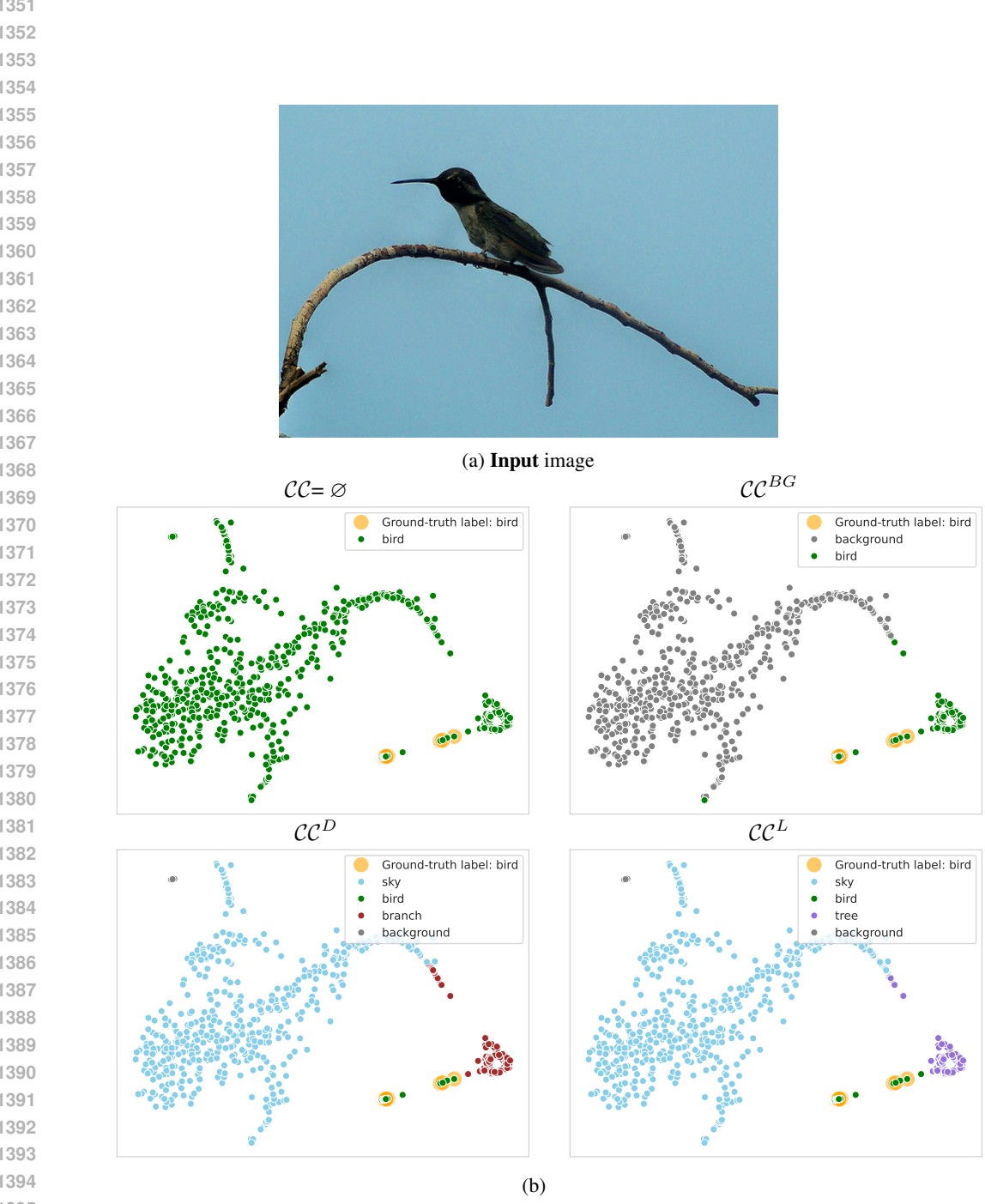

(a) **Input** image

(b)

Figure 18: **t-SNE analysis of patch features for different $\mathcal{CC}$s of an image $q$ = "bird".** We present patch features with their predicted closest text embedding coded in color. Text embeddings are corresponding $\mathcal{CC}$s of $q$ = "bird". We also mark the ground truth labels in orange. The sample is from VOC dataset.

