# OpenReview forum: "Test-time Contrastive Concepts for Open-World Semantic Segmentation"
_ICLR.cc/2025/Conference — Submitted to ICLR 2025_

### Official Review · Reviewer_1g43 · 2024-10-30

**Soundness:** 3
**Presentation:** 3
**Contribution:** 3
**Rating:** 6
**Confidence:** 4

**Summary:**

This study identified a significant issue with existing OVSS methods, which is the necessity of providing an exhaustive list of all potential visual concepts for an image to be segmented. To address this, the authors proposed a more practical scenario where only a single concept is segmented based solely on a textual prompt. To improve performance, they introduced two methods for expanding contrastive concepts: co-occurrence-based and LLM-based. Experimental results across multiple datasets demonstrate the effectiveness of their proposed methods.

**Strengths:**

1. Promising results with extensive experiments.
2. A new problem in OVSS has been identified, and a new setting of open-world open-vocabulary semantic segmentation is posed.
3. The method is simple to implement.

**Weaknesses:**

1. The problem addressed in the study lacks in-depth analysis. While the authors mention that CLIP's feature space is not easily separable, citing Miller et al.'s work, it should be noted that Miller et al.'s focus was primarily on classification and object detection tasks. Therefore, it is uncertain whether their conclusions are applicable to the image segmentation task. To provide a more comprehensive analysis, the authors should delve deeper into the feature space for the image segmentation task, comparing scenarios with and without the use of contrastive concepts.
2. The straightforward descriptions of the proposed method make it challenging to comprehend. To enhance clarity, the authors should incorporate specific examples or a conceptual diagram to illustrate their approach.
3. Comparing the proposed method directly to baselines like CAT-Seg may not be entirely fair. These baselines could potentially be enhanced through simple modifications, such as randomly selecting a subset of class names for each image during the training phase. Such adjustments could make them more suitable for the proposed setting, which involves a single query. Therefore, it would be prudent to consider such potential improvements before making direct comparisons.
4. The comparison to the classical method SAN is missing: Mengde Xu, Zheng Zhang, Fangyun Wei, Han Hu, and Xiang Bai. Side adapter network for open-vocabulary semantic segmentation. In Proceedings of the IEEE/CVF Conference on Computer Vision and Pattern Recognition, pages 2945–2954, 2023
5. To demonstrate the effectiveness of selecting/generating contrastive concepts for each query, the following two experiments are suggested to add:
    * use the full set of textual concepts $\mathcal{T}$
    * use LLM to generate many concepts (say 500) for a certain dataset
At both the training and test stages, the full set of concepts can be suplied.

**Questions:**

1. In what scenarios will the LLM-based method outperform the co-occurrence-based method? Additionally, how can one determine which method is optimal for a particular dataset?
2. Is the proposed method also suitable for multi-domain evaluation, similar to CAT-Seg?
3. Please provide some examples where the proposed method fails.

---

> ### Author Response · Authors · 2024-11-20
> **Response to Reviewer 1g43 (1/3)**
>
> We thank the reviewer for the insightful feedback and address all the concerns below. We would however like to emphasize that our method is a test-time improvement. In our comparisons, even if a method requires training, we apply our $CC$s *only at test time*. Secondly, we would like to highlight that our method considers each query concept separately at test time and generates $CC$s *per concept*. There is, therefore, no "dataset-level" adjustment of contrastive concepts $CC^{L}$ and $CC^{D}$.
>
> **W1: In-depth analysis**
>
> We thank the reviewer for the remark. Our initial statistical analysis was aligned with the findings of Miller et al., which motivated and guided our work. We however agree with the reviewer that adding such an analysis would be interesting for the community as most of the literature on CLIP space is focused on the task of image classification.
>
> In Fig.13 (appendix) of the revised version of the paper, we provide an analysis of patch-level CLIP space, using MaskCLIP features. The figure shows histograms of patch-level maximum text similarities (in cosine similarity) across 100 randomly sampled images from VOC (a) and ADE20k (b). We notice an overall concentration of cosine similarity scores in [0.1,0.3], suggesting that the feature space is not easily separable. We will integrate a discussion on separability in the paper.
>
> **W2: Conceptual diagram missing**
>
> We have added a conceptual diagram in the revised version of the paper, please see Fig. 14. We will integrate it in the final version of the paper. We also welcome any suggestions for further improvement of the figure.
>
> **W3: Comparison with supervised baselines**
>
> Our method is a test-time improvement. For CAT-Seg, which is a method supervised with mask annotations, we apply $CC$ in a zero-shot manner and show significant improvement even without retraining. Besides, trying to improve CAT-Seg and retraining it, possibly playing on candidate classes, goes beyond the use of existing methods as baselines.
>
> Tab.1 in the paper serves as a comparison between our $CC$s generation methods and was not meant to compare segmentation methods, as they use different levels of annotation. We could make this clearer in the table if the reviewer thinks it would be useful. When using our $CC^D$ and $CC^L$, compared to the baseline "background", we show significant improvements for all segmentation methods, no matter their level of supervision.
>
> **W4: Comparison to SAN**
>
> As requested by the reviewer, we run experiments with the SAN method on 2 datasets, ADE20K (presented in Tab.11) and VOC (presented in Tab.12).
>
> | Method          | $CC^{BG}$ | $CC^L$ | $CC^D$ |
> | --------------- | --------- | ------ | ------ |
> | MaskCLIP OpenAI | 20.2      | 23.5   | 25.2   |
> | DINOiser        | 28.9      | 29.7   | 31.6   |
> | CAT-Seg         | 25.7      | 38.4   | 39.7   |
> | **SAN**         | 24.5      | 35.2   | 36.1   |
> _Table 11. Results (IoU-Single%) on ADE20K including SAN._
>
> | Method          | $CC^{BG}$ | $CC^L$ | $CC^D$ |
> | --------------- | --------- | ------ | ------ |
> | MaskCLIP OpenAI | 44.2      | 52.2   | 53.4   |
> | DINOiser        | 59.3      | 63.1   | 64.7   |
> | CAT-Seg         | 52.8      | 69.5   | 67.7   |
> | **SAN**         | 50.2      | 73.4   | 73.0   |
> _Table 12. Results (IoU-Single%) on VOC including SAN._
>
> Similarly to other OVSS methods, our $CC$ generation methods applied to SAN provide consistent improvements over the $CC^{BG}$ baseline. We will include SAN results in the final version of our paper.

---

> > ### Comment · Reviewer_1g43 · 2024-11-22
> > **In-depth analysis about feature distribution**
> >
> > Thank you for your response. As for feature distribution, the current analysis is too simplified, lacking the comparison between methods with and without CC. In my opinion, using CC will make the classifier more stronger, leading to a non-linear classifier. I would like to see the comparison of feature distribution in 2D space using t-SNE with different classes shown. Without CC, it may be a simple linear classifier, which is hard to distinguish target class to various background classes. By using CC, the classifier will be more complicated, it will be easier to distinguish between the target class and the background classes. By this way, the readers can better understand why your method works.

---

> > > ### Author Response · Authors · 2024-11-25
> > > **Authors response - In-depth analysis about feature distribution**
> > >
> > > We thank the reviewer for this suggestion. We present in Fig.18 (in the newly uploaded .pdf) a t-SNE analysis over patch features from an image from the VOC dataset for the $q$ = "bird". We plot the result of classification for each separate set of $CC$s. We highlight in orange the patches that belong to the ground truth mask of class "bird". We observe that $CC^{BG}$ already helps to separate the space of background concepts from "bird" patches. However, we notice that only with $CC^L$ or $CC^D$ we can separate one visible cluster left, possibly belonging to the patch features of a branch in the image, by providing "branch" in the case of $CC^D$ or "tree" in $CC^L$. Both of our proposed methods improve the final segmentation result. We will include this discussion in the final version of our paper.

---

> > > > ### Comment · Reviewer_1g43 · 2024-11-27
> > > > **The new visualization looks better.**
> > > >
> > > > Thank you for your comment. The new visualization looks better.

---

> > > > > ### Author Response · Authors · 2024-12-03
> > > > > **Response to Reviewer 1g43**
> > > > >
> > > > > Dear reviewer, as the discussion period approaches its conclusion, we would like to thank you again for your continued engagement and insightful feedback to improve the paper. We believe we have addressed all of your concerns and we kindly ask the reviewer to re-evaluate our work.
> > > > > However, if there are any remaining questions that require further clarification, please let us know.

---

> ### Author Response · Authors · 2024-11-20
> **Response to Reviewer 1g43 (2/3)**
>
> **W5: Additional experiments**
>
> We ran the two experiments requested by the Reviewer. In Tab.13, we present results on VOC when using all concepts from LAION dictionary $T$.
>
> |       Method    | $CC^D$ | $CC^D(T)$ |
> | --------------- | ------ | --------- |
> | DINOiser        |   64.7 |      30.5 |
> | TCL             |   53.6 |      20.0 |
> | MaskCLIP OpenAI |   53.4 |      20.0 |
>
> _Table 13. Experiments on VOC with all concepts from LAION as $CC$, noted as $CC^D(T)$. Reported results are with IoU-Single%._
>
> We observe a significant drop in performance across all methods when using all $T$ as contrastive concepts.
>
> Next, we study how different numbers of LLM-generated contrastive concepts influence the results. We use CLIP-DINOiser, TCL and MaskCLIP (with OpenAI weights) models, and experiment on two datasets: Pascal VOC and Cityscapes.
>
> |   Method     | $CC^L$ | $CC^L(50)$ | $CC^L(100)$ | $CC^L(200)$ |
> | ---------:| ------:| ----------:| -----------:| -----------:|
> | DINOiser  |   63.1 |       52.9 |        50.1 |        47.9 |
> | TCL        |   52.6 |       48.3 |        45.1 |        42.0 |
> | MaskCLIP OpenAI |   52.2 |       41.4 |        39.0 |        32.2 |
>
> _Table 14. IoU-Single results on VOC dataset with $CC^L$. Experiment with LLM-generated contrastive captions ($CC^L$) for different numbers of generated concepts._
>
> |          Method  | $CC^L$ | $CC^L(50)$ | $CC^L(100)$ | $CC^L(200)$ |
> | --------------- | ------:| ----------:| ----------- | ----------- |
> | DINOiser        |   30.6 |       23.6 | 21.4        | 20.1        |
> | TCL             |   26.3 |       21.5 | 19.7        | 21.6        |
> | MaskCLIP OpenAI |   22.5 |    20.8 | 20.3   | 17.7    |
>
> _Table 15. IoU-Single results on Cityscapes dataset with $CC^L$. Experiment with LLM-generated contrastive captions ($CC^L$) for different numbers of generated concepts._
>
> Tables 14 and 15 present results for 50, 100, and 200 LLM-generated concepts. We compare them to the original setup from the paper ($CC^L$ column). We can see that using more concepts decreases performance, as measured by single-IoU across all models and datasets. We will include the ablation in the final version of the paper.
>
> **Q1: When to use $CC^L$ and when to use $CC^D$**
>
> As discussed in the main paper, we notice that $CC^L$ generalizes better to domain-specific datasets. In contrast, for both VOC and ADE20K, $CC^D$ outperforms most of the time the LLM-based $CC^L$. However, this trend does not hold for Cityscapes, where $CC^L$ gives the best results. We believe it is due to the limited availability of such driving scenes in large pretraining datasets.
>
> $CC^L$ can also yield better results for concepts that are rare or missing from the pretraining dataset. Moreover, thanks to our similarity-mapping mechanism, $CC^D$ can be used for any query concept. Yet, its effectiveness still relies on the nearest-neighbor mapping in $T$ and on the frequency of co-occurrences of $q$.
>
> It must also be kept in mind that our approach is not to be thought of for a target dataset. Indeed, if only a specific dataset is targeted, the setting is then more like a close-world scenario, and other approaches might then be more meaningful. Our approach is more appropriate for "in the wild" settings.
>
> **Q2: Multi-domain evaluation**
>
> Our single-query scenario with IoU-Single could indeed be applied to multi-domain evaluation. On the other hand, applying our $CC$ generation methods would not be as straightforward. The considered domains in the proposed multi-domain benchmark [G] span a large spectrum of very specific domains, such as *agriculture, medicine*, etc.
>
> In the case of $CC^L$, we are bounded by the capabilities of the LLM - in particular its generalization to very specific domains. In the case of $CC^D$, the contrastive concepts directly depend on the training data, which is not as specialized as the evaluation datasets.
>
> [G] Blumenstiel Benedikt, et al. "What a mess: Multi-domain evaluation of zero-shot semantic segmentation." NeurIPS 2024.

---

> > ### Author Response · Authors · 2024-11-20
> > **Response to Reviewer 1g43 (3/3)**
> >
> > **Q3: Failure case analysis**
> >
> > We present some failure cases in Fig.15 (appendix) of the revised paper. Precisely, we show examples of CLIP-DINOiser when one of the $CC$ generation methods fails.
> >
> > In the first example (first row), $CC^L$ suggests “blanket" for “bed", which typically covers the query concept. One of the potential improvements would be to instruct an LLM to also ignore potentially occluding objects. We also discuss a similar failure mode in the response to Reviewer dLMb (W1).
> >
> > In the second row, both methods fail to provide “floor" to contrast with “rug". We notice that $CC^L$ tend to be more oriented towards objects, as opposed to *stuff-like* classes. We also observe that in the $CC^L$ example, a small part of a painting on the wall is segmented as "rug". This suggests that $CC^L$ might not give a complete set of $CC$.
> >
> > Finally, in the third example, both methods fail to generate “person" to contrast with “bedclothes". However, $CC^L$ includes “pyjamas", which results in a better segmentation overall. Image-conditioned $CC$ generation (e.g., with VLMs) could be a candidate solution to that problem (please see our answer to reviewer dLMb, W1), but we leave it for future work.
> >
> > We will include this discussion in the final version of the paper.

---

### Official Review · Reviewer_ZBpw · 2024-11-02

**Soundness:** 2
**Presentation:** 2
**Contribution:** 2
**Rating:** 5
**Confidence:** 5

**Summary:**

This paper introduces a method to open-world semantic segmentation using test-time contrastive concepts (CC), addressing the challenge of segmenting specific objects in images with minimal prior information. Current Vision-Language Models (VLMs), like CLIP, assign each pixel to one of many known visual concepts. However, this method struggles when only one or a few objects are specified, especially if relevant context (e.g., related background objects) is unknown. To solve this, the authors propose generating contrastive concepts at test time, using either VLM training data or large language models (LLMs), to improve segmentation without needing extensive dataset-specific knowledge. This approach represents a step towards effective open-world segmentation by dynamically enhancing contrast without altering VLM structures or needing extensive dataset-specific adaptations. Authors evaluated the model performance on various popular benchmarks and observed consistent performance gains.

**Strengths:**

The most interesting part of this paper, in my opinion, is its deep dive into how current open-vocabulary segmentation models still lean on fixed, closed-set categories during evaluation. This is a real issue if we want these models to work effectively in real-world settings, where they’ll face plenty of unknown objects and only the object of interest is provided in the prompt.

Beside that, the paper brings some interesting contributions to the table:

1. [Test-Time Contrastive Concept Generation]: The authors introduce a fresh approach that automatically creates contrastive concepts during testing to boost segmentation in open-world scenarios. This context-specific contrast helps separate objects better without needing pre-set information.

2. [Background Handling Analysis]: They also dig into the common practice of treating "background" as a one-size-fits-all contrastive concept, showing its drawbacks and making a strong case for using concepts tailored to each query for more accurate results.

3. [New Metric for Open-World Segmentation]: To better measure performance, they propose IoU-single, a metric that assesses each concept’s segmentation independently, giving a clearer view of how well models handle open-world data.

4. [Extensive Benchmarking]: Lastly, they put their method to the test across various datasets, showing it consistently beats current open-vocabulary segmentation models, proving it’s robust and ready for broader applications.

**Weaknesses:**

1. [Problem setting] As noted in the strengths, evaluating the model's performance using only a textual prompt is a valuable contribution and can address issues faced by many existing models. However, this approach overlaps significantly with the objectives of referring segmentation and expression localization models, which aim to segment or localize objects based on a text prompt. It would be beneficial to include comparisons with existing referring segmentation or detection models to provide a clearer benchmark. While this was briefly acknowledged in the related work section, I didn't find any quantitative or qualitative comparisons. Additionally, it would be necessary to compare with works like [1], which also supports segmentation/detection of multiple instances using a single text prompt.

2. [Re-define some existing concepts] This paper labels its setting as "open-world," a term already widely used in publications, such as [2]. Redefining this established concept might lead to confusion. It may help to clarify or differentiate this setting further to avoid conflicting terminology.

3. [Paper presentation] Some tables are unclear and difficult to interpret without the context provided in the paper. For example, Table 1 lacks a clear baseline for comparison—it appears to only show different CC variants without a standard baseline to measure the gains introduced by the proposed CC method. After reading the experimental section, I understood that CC^{BG} is the baseline, but its labeling as a variant rather than a baseline is confusing. The evaluation metric is also missing from the caption, which adds to the ambiguity.


Minor Issues: There are some minor typos and grammatical errors. For example, on line 53, "without any a priori knowledge" should be "without any prior knowledge."

[1] Liu, Chang, Henghui Ding, and Xudong Jiang. "Gres: Generalized referring expression segmentation." In Proceedings of the IEEE/CVF conference on computer vision and pattern recognition, pp. 23592-23601. 2023.

[2] Sodano, Matteo, Federico Magistri, Lucas Nunes, Jens Behley, and Cyrill Stachniss. "Open-World Semantic Segmentation Including Class Similarity." In Proceedings of the IEEE/CVF Conference on Computer Vision and Pattern Recognition, pp. 3184-3194. 2024.

**Questions:**

Compared to the results of CC^{L}, the performance gains achieved by using CC^{D} appear minimal. Could this imply that simple language model-generated concepts might be sufficient for open-world segmentation? For example, in Table 1, CC^{L} can get even better results than CC^{D} on cityscapes. If this is true, then it seems like that we only need to have "hard-negative" mining to make the open-set segmentation model work?

---

> ### Author Response · Authors · 2024-11-20
> **Response to Reviewer ZBpw**
>
> **W1: Problem setting: potential overlap with goals of referring segmentation and expression localization**
>
> We thank the reviewer for this interesting remark and suggestion. As mentioned in the related work of our paper, there are substantial differences between referring segmentation and OVSS, and thus between their corresponding methods.
>
> Methods for referring segmentation, such as GRES, are highly supervised, including object relationship information, which is not the case for OVSS. While CAT-Seg has access to pixel-level class information, most OVSS methods rely solely on a model pre-trained with only image/caption annotation. Moreover, input text prompts are much more complex in referring segmentation, requiring spatial and relational reasoning, which are too complex for CLIP-like models [C], whose text encoder is limited to short texts [D].
>
> We agree, however, that methods such as GRES could be applied to OVSS. We therefore run an evaluation of GRES in a single-query scenario for OVSS on two datasets: VOC and Cityscapes. We compare GRES against CLIP-DINOiser and CAT-Seg with $CC^D$ in Tab.10.
>
> |   Method | CC type |    VOC | Cityscapes |
> | ---------- | -------:| --------:| ----------:|
> | DINOiser |  $CC^D$ |     64.7 |   **27.3** |
> | CAT-Seg  |  $CC^D$ |     67.7 |          - |
> | **GRES** |  -  | **71.2** |       14.5 |
>
> _Table 10. IoU-single% of different OVSS methods with $CC^D$, compared to GRES._
>
> On VOC, we notice that GRES outperforms the open-vocabulary segmenters even with $CC^D$. On the other hand, the results on Cityscapes point out that the reliance on strong supervision makes GRES less suitable for domain-specific datasets and for complex datasets featuring many objects. Perhaps fine-tuning on Cityscapes could address the issue.
>
> [C] Mert Yuksekgonul et al. When and why vision-language models behave like bags-of-words, and what to do about it? ICLR 2023.
>
> [D] Beichen Zhang et al. Long-CLIP: Unlocking the Long-Text Capability of CLIP. ECCV 2024.
>
> **W2: Re-defining some existing concepts: the term "open-world" is already used**
>
> We thank the reviewer for this important remark. It is indeed crucial to avoid potential confusion on terminology.
>
> The original "open-world" setting proposed in [E] is a pragmatic approach for applications - requiring models to identify novel (unknown, out-of-distribution (OOD)) classes, have them progressively labelled and further learned incrementally. In recent years, the novel class identification issue has been addressed with VLM-based zero-shot, open-vocabulary or OOD methods [F].
>
> Here, we also aim for a practical and realistic scenario, going beyond the typical open-vocabulary benchmarks that still assume access to a pre-defined and static list of dataset-specific classes, which is ultimately a closed-world setup. With "open-world", we want to highlight this difference, stressing that our contrastive concepts are automatically suggested for each new user query, with concepts not tied to any specific dataset or benchmark. (When we evaluate on a benchmark, we do not assume that the list of classes appearing in the benchmark is known). In contrast, just saying "open-vocabulary" could offer freedom in the formulation of concepts while nevertheless restricting to a domain, which we want to exclude here.
>
> We propose to rename our task "*open-class*" segmentation, as the goal of our work is to break the dataset-level dependencies and to consider classes as independent concepts. We are also gladly open to other suggestions.
>
> [E] Abhijit Bendale \& Terrance Boult. Towards Open World Recognition. CVPR 2015.
>
> [F] Jianzong Wu et al. Towards open vocabulary learning: A survey. T-PAMI 2024.
>
> **W3: Paper presentation**
>
> We apologize for the confusion and provide a revised version of Tab.1 in the attached revision of our paper. We also corrected the mentioned sentence and removed any typos we could find.
>
> **Q1: $CC^L$ sufficient for open-world segmentation?**
>
> As discussed in Sec. 4.3 of our paper, we found $CC^L$ more suitable for "domain-specific" datasets, such as Cityscapes. We believe this is due to the limited availability of driving scenes in large pretraining datasets. On the other hand, on "general-domain" images such as images in ADE20K, $CC^D$ consistently outperforms $CC^L$. This suggests that each approach has specific merits, which may benefit one setting more than another. We also observe in the qualitative examples in Fig.4 that multiple contrastive concepts (columns $CC^D(all)$ and $CC^L(all)$) can be useful for good segmentation.
>
> As the reviewer phrases it, $CC^L$ can indeed be thought of as a way to find "hard" negatives, i.e., the most relevant concepts to contrast with. But $CC^D$ is also a way to do so, relying on CLIP training-time concepts rather than LLM-based open concepts. Finding multiple good negatives is exactly the motivation for our Contrastive Concepts.
> In case we misunderstood the question, we kindly ask the reviewer to let us know.

---

> > ### Author Response · Authors · 2024-11-25
> > **Request by the authors**
> >
> > Dear reviewer, as the end of the discussion period approaches, we kindly ask for feedback regarding our reply. Should there be any remaining concerns, we will gladly address them.

---

> > > ### Comment · Reviewer_ZBpw · 2024-11-29
> > >
> > > Hi authors,
> > >
> > > Thank you for your detailed responses. Most of my questions have been addressed, and I appreciate the effort in clarifying your approach. By leveraging the text distribution in the VLM’s training set or carefully crafted LLM prompts, the authors propose methods to automatically generate textural contrastive concepts to replace the generic "background" label. While I agree after the rebuttal that this method can indeed improve model performance on popular benchmarks, I still have concerns about the technical contributions and practical implications of the proposed approach—particularly regarding the use of LLMs to define a new label space during testing.
> > >
> > > These concerns are, of course, my personal and somewhat subjective perspectives.
> > >
> > > 1) The label space is a critical factor in open-set semantic segmentation, but it is not something that can or should be predefined, especially in real-world applications. The label space depends heavily on the downstream task or user-specific requirements rather than being tied to a single image or a specific class name. This inherent dependence on task and user context makes predefined label spaces less flexible and limits their applicability across diverse scenarios.
> > > My current rating, borderline reject, reflects my philosophical concerns about how to define background and foreground in open-set segmentation. While relying on an LLM to provide a new background label space may boost performance on certain benchmarks, this approach risks lacking the flexibility required for various downstream applications.
> > >
> > > 2) Additionally, the reliance on a VLM to enhance the performance of a segmentation model raises practical concerns. This approach essentially uses a heavyweight model to improve a lightweight segmentation model, which could undermine real-time performance—a critical requirement for many segmentation and detection tasks. Incorporating a VLM to dynamically redefine the "label space" during each run may significantly slow down processing, making it less feasible for real-time applications.
> > >
> > > Thank you again for your thoughtful responses, and I hope my feedback helps refine the paper and future directions of this work.

---

> > > > ### Author Response · Authors · 2024-11-30
> > > > **Response to reviewer ZBpw**
> > > >
> > > > 1. Yes, the label space is critical and should definitely not be predefined. We propose two methods:
> > > > $CC^D$ tries to find the best contrasts given the "label space" that the underlying segmenter (OVSS) was trained on. In other words, it is only bounded by what the OVSS can segment. If there were better contrasting concepts, the OVSS could not exploit them anyway.
> > > > $CC^L$ is unbounded as the LLM that we query can potentially generate completely arbitrary text.
> > > > Therefore, in no way are our two approaches limiting the "label space".
> > > >
> > > >
> > > > 2. We assume we are given an OVSS, which does the core segmentation work. Then regarding performance:
> > > > - $CC^D$ runs in a few milliseconds.
> > > > - $CC^L$ has to query an LLM, which may take a few seconds.
> > > >
> > > > Please note that what we propose does not rely on a VLM in any way (besides the underlying OVSS). Our additional experiment using a VLM in the rebuttal was an experiment suggested by Reviewer dLMb, but it is not a part of our current proposal.
> > > > Therefore, we believe our approach is practical for real-time/interactive scenarios, where a user simply wants to segment something particular in a given image. And even if $CC^L$ is too slow for such a user, then $CC^D$ definitely will not be.

---

> > > > > ### Author Response · Authors · 2024-12-03
> > > > > **Response to reviewer ZBpw**
> > > > >
> > > > > Dear reviewer, as the discussion period approaches its conclusion, we would like to thank you again for your continued engagement and insightful feedback to improve the paper. We hope that our previous answer addresses your concerns.
> > > > > If there are any remaining questions that require further clarification, please let us know.

---

### Official Review · Reviewer_dLMb · 2024-11-04

**Soundness:** 3
**Presentation:** 3
**Contribution:** 3
**Rating:** 6
**Confidence:** 5

**Summary:**

The paper discussed the 'background' concept in open-world semantic segmentation and proposed a method to suppress background with contrastive concepts when segmenting a specific category in test time. In addition, a new single-query evaluation setup for open-world semantic segmentation that does not rely on any domain knowledge is proposed, and a new metric is designed to evaluate the grounding of visual concepts. I think the "background" is confusing in segmentation but is not well-discussed and well-defined. It is valuable to study this problem.

**Strengths:**

1. Automatic discovery of comparison concepts is valuable because artificially defining comparison concepts requires professional knowledge and a manual preset knowledge base in advance.
2. I agree that the concept of "background" is confusing in segmentation but is not well discussed. Therefore, the article's exploration of "background" is valuable. This paper discussed the CLIP-based method, in which the pixel is scored by cosine similarity, and the threshold filters the background. However, thresholds are difficult to set and often overfit specific datasets, which is a real pain point.
3. Experiments on multiple datasets and verification of multiple methods prove the effectiveness of the proposed method.

**Weaknesses:**

1. When using LLM for contrastive concepts extraction, do you consider generating corresponding contrastive concepts for each image (such as inputting the image along with the category name), and whether this is better than just asking with text? (For example, if you type a boat and LLM will tell you water, person, beach), but if a boat is stranded on the beach and you enter the image together, LLM(or VLM) will only output 'beach', which is more targeted when you divide it later.
2. How will this method be handled when the same pixels (or region) belong to the whole and parts (such as person and arm) at the same time? Because for ‘arm’, the rest region of the ’person‘ is the background. Can you give an example to describe the process briefly?
2. Can the background class be circumvented if the similarity is calculated using ‘sigmoid’ instead of ‘softmax’? You can have a try.

**Questions:**

See weakness part.

---

> ### Author Response · Authors · 2024-11-20
> **Response to Reviewer dLMb (1/2)**
>
> **W1: What about image-based contrastive concepts?**
>
> Our constrastive concepts are currently only based on the given query, not on the image. Using the image as well, which contains more specific information, is definitely an interesting research direction. Thank you for the suggestion.
> To try the idea, we experiment with LLAVA-1.5 7B [A] on VOC and present the results in Tab.9. As suggested by the reviewer, we prompt LLAVA using both $q$ and an image. We instruct the model to generate a list of visual concepts surrounding $q$ on the given image. As for LLM prompting, we ask the model explicitly not to return synonyms and parts of $q$. We notice, however, that the model has trouble following the instructions, quite frequently returning an empty list. The model also does not fully follow the requirement of ignoring synonyms.
>
> | Method   | $CC^{BG}$ | $CC^L$ | $CC^D$ | $CC^{VLM}$ |
> | -------- | --------- | ------ | ------ | ---------- |
> | MaskCLIP | 44.2      | 52.2   | 53.4   | 47.7       |
> | DINOiser | 59.3      | 63.1   | 64.7   | 61.9       |
>
> _Table 9. IoU-single% of different OVSS methods on VOC. We note $CC^{VLM}$ the VLM-based contrastive concept setting described in the text._
>
> Our results show that the VLM-based approach underperforms $CC^L$ and $CC^D$. We attribute it to the fact that our prompt requires additional steps of reasoning (removing synonyms, etc.) for which the language model in the considered VLM can be too weak. We believe it is still a promising direction which we leave for future work. One might consider improving the prompt, using a VLM with stronger language and reasoning capabilities, or using both a VLM and an LLM, etc.
>
> [A] Haotian Liu et al. Improved baselines with visual instruction tuning. CVPR 2024.
>
> **W2: What happens when querying the part of an object?**
>
> This is an interesting question. We reuse here the proposed example where we query alternatively the whole object "person" or the part "arm". We start by discussing a single-query scenario, which is the main focus of our work.
>
> In the case of the query $q$ being "parent class", such as "person" in the given example, we have the following mechanism:
> - For $CC^L$, we explicitly encode in the prompt an instruction not to include parts of query objects (please see Fig.8 of our paper).
> - For $CC^D$, we also considered an explicit part removal, but we found it was not impacting the overall performance on tested datasets (see Sec. C.3 of our appendix).
>
> In the case of $q$ being a part of a parent object ("arm" in our considered example), we did not design any specific mechanism, therefore if both are present the method is bounded by the segmentation method itself and its feature separability.
> We qualitatively investigate such an example in Fig. 16 of our revised pdf. We present the results when $q$="arm" (top row) and $q$="person" (bottom row) for both $CC^D$ and $CC^L$.
> - In the case of "person", as described above, none of the $CC$ interfere with $q$.
> - In the case of "arm", we notice that both methods produce $CC$ that interfere with $q$, $CC^D$ includes "coat" which covers part of "arm", leaving only the hand properly segmented. In $CC^L$, "sleeve" unfortunately covers the whole "arm".
>
> Please note however that there is an intrinsic ambiguity in the task regarding the interpretation of queries. For instance, if part of a person is occluded by an object, the occluded part should not be segmented. On the contrary, if "occluded" by clothes, the clothes should probably be segmented along with the visible parts of the body.
>
> A potential improvement could be to incorporate another filtering mechanism which removes potential occluding concepts. Another alternative would be to consider the top $K$ concepts and allow for multi-label segmentation. We believe that considering hierarchies is an important direction for future work.
>
> When considering both prompts at once, we would follow our multi-query process described in Sec 3.1. and combine $CC$s from both queries with the filtering mechanism. With our filtering mechanism, we ensure that $CC_{q=arm}$ does not interfere with $q=$"person" and the other way around. We however emphasize that the final segmentation quality of "arm" vs "person" is dependent on the CLIP space. A final segmentation result is thus determined by the clip-similarity score, meaning that the closest concept wins. In the case of our example, this would be "person".

---

> > ### Author Response · Authors · 2024-11-20
> > **Response to Reviewer dLMb (2/2)**
> >
> > **W3: Trying sigmoid instead of softmax for similarity**
> >
> > Trying to separate a query from its background using a binary criterion is a natural alternative direction to consider, and using a sigmoid to make such a decision makes a lot of sense.
> >
> > We test using a sigmoid on CLIP similarity scores. We show the results of an experiment with CLIP-DINOiser on VOC in Fig. 17 (appendix) of our revised pdf file. We make the following observations: (1) none of the thresholds allow us to reach the performance of $CC^{BG}$, and (2) the performance is very sensitive to the value of the threshold. We believe this is due to the CLIP space not being easily separable, with most similarity scores in the [0.1,0.3] value range (see Fig.13 of our revised version of the paper). We discuss this difficulty in section 3.1 (L198-L202) of the paper.
> >
> > Alternatively, one could use SigLIP [B] as a backbone, which is trained with a Sigmoid loss and therefore suitable for such inference. However, to the best of our knowledge, most OVSS methods still use CLIP (trained with softmax) as a backbone. We keep this direction in future work.
> >
> > [B] Xiaohua Zhai et al. Sigmoid loss for language image pre-training. ICCV 2023.

---

> > > ### Comment · Reviewer_dLMb · 2024-11-25
> > >
> > > I am interested in the performance when you use SigLIP [B] with "sigmoid"?  Because in real-world scenarios, the category is unknown. The score of softmax is related to the given category vocabulary, and changing the vocabulary will result in different classification scores. So I believe that sigmoid-based classification will be more reasonable in open scenarios  in the future.

---

> ### Author Response · Authors · 2024-11-29
> **SigLIP experiments**
>
> Following the reviewer's request, we experiment with SigLIP ViT-B/16 backbone. We present results on the datasets VOC in Tab. 16, ADE20k in Tab. 17 and Cityscapes in Tab. 18.  We report results using our metric IoU-Single when employing different $CC$s, as well as when using a non-contrastive Sigmoid-based thresholding strategy (noted **Sigmoid**) for SigLIP. We consider different thresholds which we observe lead to a high variability in results. We compare the results with those of MaskCLIP with different $CC$s as is it the closest method which does not include any refinement mechanism. Overall, the results obtained with SigLIP (either using $CC$ or sigmoid) are significantly worse than those obtained with MaskCLIP on VOC. On ADE20k and Cityscapes, SigLIP with sigmoid obtains closer results to MaskCLIP; however, it is always outperformed by MaskCLIP when using $CC^D$ and/or $CC^{L}$. Additionally, we observe that the sigmoid results are highly sensitive to the threshold, with a drop of between -4 and -7 pts (eg. from 20.6 to 13.8/12.1 on Cityscapes) when the delta changes by *as little as 0.005*. We will integrate this discussion in the final paper, as we believe it is insightful.
>
> | Method | $CC^{BG}$ | $CC^L$ | $CC^D$ | **Sigmoid (th=0.5)** | **Sigmoid (th=0.505)** | **Sigmoid (th= 0.51)** |
> | ------ | --------- | ------ | ------ | ------------------ | -------------------- | ------------------- |
> | SigLIP | 26.9      | 27.4   | 27.2   | 28.9               | 33.1                 | 29.3                |
> | MaskCLIP OpenAI |    44.2  |   52.2 | **53.4**|  -   | - |-
>
> _Table 16. Results (IoU-Single%) on VOC with SigLIP backbone._
>
> | Method | $CC^{BG}$ | $CC^L$ | $CC^D$ | **Sigmoid (th=0.5)** | **Sigmoid (th=0.505)** | **Sigmoid (th= 0.51)** |
> | ------ | --------- | ------ | ------ | ------------------ | -------------------- | ------------------- |
> | SigLIP | 9.8       | 20.7   | 21.8   | 20.0                  | 24.1                 | 18.1                |
> | MaskCLIP OpenAI | 20.2      | 23.5   | **25.2**   | - | - | -
>
> _Table 17. Results (IoU-Single%) on ADE20k with SigLIP backbone._
>
> | Method | $CC^{BG}$ | $CC^L$ | $CC^D$ | **Sigmoid (th=0.5)** | **Sigmoid (th=0.505)** | **Sigmoid (th= 0.51)** |
> | ------ | --------- | ------ | ------ | ------------------ | -------------------- | ------------------- |
> | SigLIP | 6.0       | 19.3  | 11.4  |             13.8      |     20.6            |        12.1         |
> | MaskCLIP OpenAI | 15.0     | **22.5**   | 22.0  |  - | - | -
>
> _Table 18. Results (IoU-Single%) on Cityscapes with SigLIP backbone._

---

> > ### Author Response · Authors · 2024-12-03
> > **Response to Reviewer dLMb**
> >
> > Dear reviewer, as the discussion period approaches its conclusion, we would like to thank you again for your continued engagement and insightful feedback to improve the paper. We believe we have addressed all of your concerns and we kindly ask the reviewer to re-evaluate our work.
> > However, if there are any remaining questions that require further clarification, please let us know.

---

### Author Response · Authors · 2024-11-20
**General comment**

We thank the reviewers for their insightful feedback and their appreciation of our work and contributions. We clarify the concerns in separate responses below. We also updated the pdf file of our paper to include the necessary visual materials supporting our discussion (at the end of the Appendix). The new or updated content is highlighted in blue and referenced in the responses here below. We kindly ask the reviewers to check the corresponding sections mentioned in our responses.

---

### Meta-Review · Area_Chair_mUGP · 2024-12-23

**Metareview:**

This paper introduces a test-time method for open-vocabulary semantic segmentation (OVSS) that suppresses false positive segments using contrastive concepts (CC) when segmenting a specific category in test time. The authors propose generating contrastive concepts at test time using either VLM training data or large language models (LLMs), which can be applied to off-the-shelf OVSS methods to improve performance. In addition, a new single-query evaluation setup and a new metric are also proposed to evaluate the grounding of visual concepts.

While all reviewers appreciated the authors’ investigation into the issue of requiring pre-set information on all potential visual concepts in OVSS, they also raised concerns about the limited technical contribution, lack of in-depth analysis, missing comparisons, and unclear presentations, giving borderline scores. After discussion, some issues were addressed, but no reviewers strongly supported this paper, retaining borderline scores. The main issue, among others, is the limited technical contribution; reviewer dLMb pointed out that enabling the sigmoid-based classification would be more reasonable and generalizable in open scenarios rather than using the proposed test-time trick, and reviewer ZBpw had concerns that relying on LLM or the information of VLM training set may not be generalizable in many practical downstream applications. After carefully reading the manuscript, the reviews, and the discussions, AC agreed that the technical contribution of this paper is limited; the proposed method suppresses false positive predictions as post-processing in test time using train-set information or LLM rather than making the predictor itself more robust to background clutter. In particular, such background suppression/filtering tricks have been suggested earlier, e.g., in (Wysoczanska et al., 2024a;b), and the proposed method does not significantly outperform them; e.g., the saliency method (Wysoczanska et al., 2024b) shows comparable or better results in Table 2 and Table 7. It is unclear what the merit of the proposed method is compared to such alternatives.

**Additional Comments On Reviewer Discussion:**

After rebuttal and discussion, all reviewers retained their original borderline scores (two borderline accepts and one borderline reject). The concerns were the limited technical contribution, lack of in-depth analysis, missing comparisons, and unclear presentations. After the discussion, some issues were addressed, but the concerns about the limited technical contribution remained, and AC agreed with them. See above.

---

### Decision · Program_Chairs · 2025-01-22

Reject